# GENERATIVE SLICED MMD FLOWS WITH RIESZ KERNELS

**Johannes Hertrich**[1], **Christian Wald**[2], **Fabian Altekrüger**[3], **Paul Hagemann**[2]
[1] University College London, [2] Technische Universität Berlin, [3] Humboldt-Universität zu Berlin
Correspondence to: `j.hertrich@ucl.ac.uk`

## ABSTRACT

Maximum mean discrepancy (MMD) flows suffer from high computational costs in large scale computations. In this paper, we show that MMD flows with Riesz kernels $K(x,y) = -\|x-y\|^r$, $r \in (0,2)$ have exceptional properties which allow their efficient computation. We prove that the MMD of Riesz kernels, which is also known as energy distance, coincides with the MMD of their sliced version. As a consequence, the computation of gradients of MMDs can be performed in the one-dimensional setting. Here, for $r = 1$, a simple sorting algorithm can be applied to reduce the complexity from $O(MN + N^2)$ to $O((M+N)\log(M+N))$ for two measures with $M$ and $N$ support points. As another interesting follow-up result, the MMD of compactly supported measures can be estimated from above and below by the Wasserstein-1 distance. For the implementations we approximate the gradient of the sliced MMD by using only a finite number $P$ of slices. We show that the resulting error has complexity $O(\sqrt{d/P})$, where $d$ is the data dimension. These results enable us to train generative models by approximating MMD gradient flows by neural networks even for image applications. We demonstrate the efficiency of our model by image generation on MNIST, FashionMNIST and CIFAR10.

## 1 INTRODUCTION

With the rise of generative models, the field of gradient flows in measure spaces received increasing attention. Based on classical Markov chain Monte Carlo methods, Welling & Teh (2011) proposed to apply the Langevin dynamics for inferring samples from a known probability density function. This corresponds to simulating a Wasserstein gradient flow with respect to the Kullback-Leibler divergence, see Jordan et al. (1998). Closely related to this approach are current state-of-the-art image generation methods like score-based models (Song & Ermon, 2019; 2020) or diffusion models (Ho et al., 2020; Song et al., 2021), which significantly outperform classical generative models like GANs (Goodfellow et al., 2014) or VAEs (Kingma & Welling, 2014). A general aim of such algorithms (Arbel et al., 2021; Ho et al., 2020; Wu et al., 2020) is to establish a path between input and target distribution, where "unseen" data points are established via the randomness of the input distribution. Several combinations of such Langevin-type Markov chain Monte Carlo methods with other generative models were proposed in (Ben-Hamu et al., 2022; Hagemann et al., 2023; Wu et al., 2020). Gradient flows on measure spaces with respect to other metrics are considered in (di Langosco et al., 2022; Dong et al., 2023; Grathwohl et al., 2020; Liu, 2017; Liu & Wang, 2016) under the name Stein variational gradient descent.

For approximating gradient flows with respect to other functionals than the KL divergence, the authors of (Altekrüger et al., 2023; Ansari et al., 2021; Alvarez-Melis et al., 2022; Bunne et al., 2022; Fan et al., 2022; Gao et al., 2019; Garcia Trillos et al., 2023; Heng et al., 2023; Mokrov et al., 2021; Peyré, 2015) proposed the use of suitable forward and backward discretizations. To reduce the computational effort of evaluating distance measures on high-dimensional probability distributions, the sliced Wasserstein metric was introduced in (Rabin et al., 2012). The main idea of the sliced Wasserstein distance is to compare one-dimensional projections of the corresponding probability distributions instead of the distributions themselves. This approach can be generalized to more general probability metrics (Kolouri et al., 2022) and was applied in the context of Wasserstein gradient flows in (Bonet et al., 2022b; Liutkus et al., 2019).

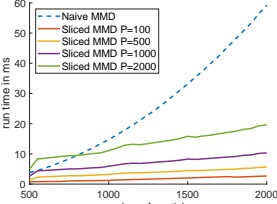 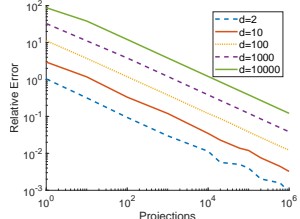 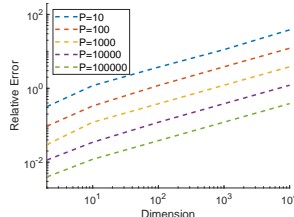

Figure 1: Left: Comparison of run time for 1000 gradient evaluations of naive MMD and sliced MMD with different number of projections $P$ in the case $d = 100$. Middle and right: Relative error of the gradients of sliced MMD and MMD with respect to the number $P$ of projections and the dimension $d$. The results show the relative error behaves asymptotically as $O(\sqrt{d/P})$ as shown in Theorem 4.

For many generative gradient-flow methods it is required that the considered functional can be evaluated based on samples. For divergence-based functionals like the Kullback-Leibler or the Jensen-Shannon divergence, a variational formulation leading to a GAN-like evaluation procedure is provided in (Fan et al., 2022). In contrast, the authors of (Altekrüger et al., 2023; Arbel et al., 2019; Glaser et al., 2021) use functionals based on the maximum mean discrepancy (MMD) which can be directly evaluated based on empirical measures. For positive definite kernels, it can be shown under some additional assumptions that MMD defines a metric on the space of probability distributions, see e.g., (Gretton et al., 2012; Sriperumbudur et al., 2011; 2010). If the considered kernel is smooth, then Arbel et al. (2019) proved that Wasserstein gradient flows can be fully described by particles. Even though this is no longer true for non-smooth kernels (Hertrich et al., 2023b), Altekrüger et al. (2023) pointed out that particle flows are Wasserstein gradient flows at least with respect to a restricted functional. In particular, we can expect that particle flows provide an accurate approximation of Wasserstein gradient flows as long as the number of particles is large enough.

**Contributions.** The computational complexity of MMD between two empirical measures with $N$ and $M$ support points depends quadratically on $N$ and $M$, which makes large scale computations impossible. In this paper, we focus on the MMD with *Riesz kernels*

$$K(x, y) = -\|x - y\|^r, \quad r \in (0, 2), \tag{1}$$

also known as energy distance (Sejdinovic et al., 2013; Székely, 2002; Székely & Rizzo, 2013). We show that Riesz kernels have the outstanding property that their MMD coincides with the sliced MMD of univariate Riesz kernels. It is this property that enables us to reduce the computation of (gradients of) MMD to the one-dimensional setting. In the case of $r = 1$, we propose a simple and computationally very efficient sorting algorithm for computing the gradient of the one-dimensional MMD with complexity $O((M + N) \log(M + N))$. Considering that our numerical examples use between 10.000 and 50.000 particles, this leads to an incredible speed-up for gradient computations of (sliced) MMD as illustrated in the left plot of Figure 1. Our approach opens the door to applications in image processing, where we have usually to cope with high dimensional data.

In practice, sliced probability metrics are evaluated by replacing the expectation over all projections by the empirical expectation resulting in a finite sum. In the case of sliced MMD with Riesz kernels and $r = 1$, we prove that the error induced by this approximation behaves asymptotically as $O(\sqrt{d/P})$, where $d$ is the data dimension and $P$ the number of projections, see the middle plot in Figure 1 for an illustration. The square root scaling of the error in the dimension $d$ ensures that an accurate computation of the sliced MMD with Riesz kernels is possible even in very high dimensions as demonstrated in the right plot in Figure 1. Taking into account the number of projections, the overall complexity of the computation of the derivatives of MMD is $O(dP(M + N) \log(M + N))$.

We apply the cheap evaluation of MMD gradients to compute MMD particle flows starting with samples from an initial probability measure $\mu_0$ to samples from a predefined target distribution $\nu$, which is given by samples. Finally, we derive a generative model by training a sequence $(\Phi_l)_{l=1}^{L}$ of neural networks, where each $\Phi_l$ approximates a certain number of steps of the particle flow. This allows us to train our network iteratively. In particular, during the training and evaluation procedure, we always consider only one of the networks $\Phi_l$ at the same time. This allows an efficient training with relatively low resources even though all networks $\Phi_l$ together have a large number of parameters.

We demonstrate the efficiency of our generative sliced MMD flows for image generation on MNIST, FashionMNIST and CIFAR10.

**Related Work.** Gradient flows with respect to MMD functionals are considered in (Altekrüger et al., 2023; Arbel et al., 2019; Hertrich et al., 2023b; Kolouri et al., 2022). However, due to the quadratic complexity of the computation of the derivative of MMD functionals in the number of support points of the involved measures, these papers have a rather theoretical scope and applications are limited to measures supported on a few hundred points. In order to reduce the dimension of the problem, Kolouri et al. (2022) consider a sliced version of MMD. This is motivated by the success of sliced Wasserstein distances (Rabin et al., 2012), which were used for deriving gradient flows in (Bonet et al., 2022b; Liutkus et al., 2019; Nguyen et al., 2023; 2021). In particular, Kolouri et al. (2022) observe that the sliced MMD is again a MMD functional *with a different kernel*. We use this result in Section 2. Vayer & Gribonval (2023) bound Wasserstein distances and MMD against each other. However, they use strong assumptions on the kernel, which are not fulfilled for the negative distance kernel. In very low dimensions, fast evaluations of MMD and their gradients were proposed in (Gräf et al., 2012; Teuber et al., 2011) based on fast Fourier summation using the non-equispaced fast Fourier transforms (NFFT), see (Plonka et al., 2018, Sec. 7) and references therein. Unfortunately, since the complexity of the NFFT depends exponentially on the data-dimension, these approaches are limited to applications in dimension four or smaller. In a one-dimensional setting, the energy distance is related to the Cramer distance, see (Székely, 2002). In the context of reinforcement learning, Lhéritier & Bondoux (2021) developed fast evaluation algorithms for the latter based on the calculation of cumulative distribution functions.

Finally, the authors of (Bińkowski et al., 2018; Dziugaite et al., 2015; Li et al., 2017; 2015) apply MMD for generative modelling by constructing so-called MMD-GANs. However, this is conceptionally a very different approach since in MMD-GANs the discriminator in the classical GAN framework (Goodfellow et al., 2014) is replaced by a MMD distance with a variable kernel. Also relevant is the direction of Sobolev-GANs (Mroueh et al., 2018) in which the discriminator is optimized in a Sobolev space, which is related to the RKHS of the Riesz kernel. Similar to GAN ideas this results in a max-min problem which is solved in an alternating fashion and is not related to gradient flows.

**Outline of the Paper.** In Section 2, we prove that the sliced MMD with the one-dimensional Riesz kernel coincides with MMD of the scaled $d$-dimensional kernel. This can be used to establish an interesting lower bound on the MMD by the Wasserstein-1 distance. Then, in Section 3 we propose a sorting algorithm for computing the derivative of the sliced MMD in an efficient way. We apply the fast evaluation of MMD gradients to simulate MMD flows and to derive a generative model in Section 4 . Section 5 shows numerical experiments on image generation. Finally, conlusions are drawn in Section 6. The appendices contain the proofs and supplementary material.

## 2    SLICED MMD FOR RIESZ KERNELS

Let $\mathcal{P}(\mathbb{R}^d)$ denote the set of probability measures on $\mathbb{R}^d$ and $\mathcal{P}_p(\mathbb{R}^d)$ its subset of measures with finite $p$-th moment, i.e., $\int_{\mathbb{R}^d} \|x\|^p \mathrm{d}\mu(x) < \infty$. Here $\|\cdot\|$ denotes the Euclidean norm on $\mathbb{R}^d$. For a symmetric, positive definite kernel $K\colon \mathbb{R}^d \times \mathbb{R}^d \to \mathbb{R}$, the *maximum mean discrepancy* (MMD) $\mathcal{D}_K\colon \mathcal{P}(\mathbb{R}^d) \times \mathcal{P}(\mathbb{R}^d) \to \mathbb{R}$ is the square root of $\mathcal{D}_K^2(\mu,\nu) \coloneqq \mathcal{E}_K(\mu-\nu)$, where $\mathcal{E}_K$ is the *interaction energy* of signed measures on $\mathbb{R}^d$ defined by

$$\mathcal{E}_K(\eta) \coloneqq \frac{1}{2} \int_{\mathbb{R}^d} \int_{\mathbb{R}^d} K(x,y)\, \mathrm{d}\eta(x)\mathrm{d}\eta(y).$$

Due to its favorable properties, see Appendix E, we are interested in Riesz kernels

$$K(x,y) = -\|x-y\|^r, \quad r \in (0,2).$$

These kernels are only conditionally positive definite, but can be extended to positive definite kernels by $\tilde{K}(x,y) = K(x,y) - K(x,0) - K(0,y)$, see also Remark 13. Then it holds for $\mu, \nu \in \mathcal{P}_r(\mathbb{R}^d)$ that $\mathcal{D}_K(\mu,\nu) = \mathcal{D}_{\tilde{K}}(\mu,\nu)$, see (Neumayer & Steidl, 2021, Lemma 3.3 iii)). Moreover, for Riesz kernels, $\mathcal{D}_K$ is a metric on $\mathcal{P}_r(\mathbb{R}^d)$, which is also known as so-called energy distance (Sejdinovic et al., 2013; Székely & Rizzo, 2013). Note that we exclude the case $r = 2$, since $\mathcal{D}_K$ is no longer a metric in this case.

However, computing MMDs on high dimensional spaces is computationally costly. Therefore, the *sliced MMD* $\mathcal{SD}^2_{\mathrm{k}} \colon \mathcal{P}_2(\mathbb{R}^d) \times \mathcal{P}_2(\mathbb{R}^d) \to \mathbb{R}$ was considered in the literature, see e.g., Kolouri et al. (2022). For a symmetric 1D kernel $\mathrm{k} \colon \mathbb{R} \times \mathbb{R} \to \mathbb{R}$ it is given by

$$\mathcal{SD}^2_{\mathrm{k}}(\mu,\nu) \coloneqq \mathbb{E}_{\xi \sim \mathcal{U}_{\mathbb{S}^{d-1}}}[\mathcal{D}^2_{\mathrm{k}}(P_{\xi \#}\mu, P_{\xi \#}\nu)]$$

with the push-forward measure $P_{\xi \#}\mu \coloneqq \mu \circ P_\xi^{-1}$ of the projection $P_\xi(x) \coloneqq \langle \xi, x \rangle$ and the uniform distribution $\mathcal{U}_{\mathbb{S}^{d-1}}$ on the sphere $\mathbb{S}^{d-1}$. By interchanging the integrals from the expectation and the definition of MMD, Kolouri et al. (2022) observed that the sliced MMD is equal to the MMD with an associate kernel $\mathrm{K} \colon \mathbb{R}^d \times \mathbb{R}^d \to \mathbb{R}$. More precisely, it holds

$$\mathcal{SD}^2_{\mathrm{k}}(\mu,\nu) = \mathcal{D}^2_{\mathrm{K}}(\mu,\nu), \quad \text{with} \quad \mathrm{K}(x,y) \coloneqq \mathbb{E}_{\xi \sim \mathcal{U}_{\mathbb{S}^{d-1}}}[\mathrm{k}(P_\xi(x), P_\xi(y))].$$

By the following theorem, this relation becomes more simple when dealing with Riesz kernels, since in this case the associate kernel is a Riesz kernel as well.

**Theorem 1** (Sliced Riesz Kernels are Riesz Kernels). *Let $\mathrm{k}(x,y) \coloneqq -|x-y|^r$, $r \in (0,2)$. Then, for $\mu, \nu \in \mathcal{P}_r(\mathbb{R}^d)$, it holds $\mathcal{SD}^2_{\mathrm{k}}(\mu,\nu) = \mathcal{D}^2_{\mathrm{K}}(\mu,\nu)$ with the associated scaled Riesz kernel*

$$\mathrm{K}(x,y) \coloneqq -c_{d,r}^{-1}\|x-y\|^r, \quad c_{d,r} \coloneqq \frac{\sqrt{\pi}\Gamma(\frac{d+r}{2})}{\Gamma(\frac{d}{2})\Gamma(\frac{r+1}{2})}.$$

The proof is given in Appendix A. The constant $c_{d,r}$ depends asymptotically with $O(d^{r/2})$ on the dimension. In particular, it should be "harder" to estimate the MMD or its gradients in higher dimensions via slicing. We will discuss this issue more formally later in Remark 5. For $r = 1$, we just write $c_d \coloneqq c_{d,1}$ and can consider measures in $\mathcal{P}_1(\mathbb{R}^d)$. Interestingly, based on Theorem 1, we can establish a relation between the MMD and the Wasserstein-1 distance on $\mathcal{P}_1(\mathbb{R}^d)$ defined by

$$\mathcal{W}_1(\mu,\nu) \coloneqq \min_{\pi \in \Pi(\mu,\nu)} \int \|x-y\| \, \mathrm{d}\pi(x,y),$$

where $\Pi(\mu,\nu)$ denotes the set of measures in $\mathcal{P}_1(\mathbb{R}^d \times \mathbb{R}^d)$ with marginals $\mu$ and $\nu$. This also shows that Conjecture 1 in (Modeste & Dombry, 2023) can only hold for non-compactly supported measures. The proof is given in Appendix B.

**Theorem 2** (Relation between $\mathcal{D}_K$ and $\mathcal{W}_1$ for Distance Kernels). *Let $K(x,y) \coloneqq -\|x-y\|$. Then, it holds for $\mu, \nu \in \mathcal{P}_1(\mathbb{R}^d)$ that $2\mathcal{D}^2_K(\mu,\nu) \leq \mathcal{W}_1(\mu,\nu)$. If $\mu$ and $\nu$ are additionally supported on the ball $B_R(0)$, then there exists a constant $C_d > 0$ such that $\mathcal{W}_1(\mu,\nu) \leq C_d R^{\frac{2d+1}{2d+2}} \mathcal{D}_K(\mu,\nu)^{\frac{1}{d+1}}$.*

The fact that the sample complexities of MMD and Wasserstein-1 are $O(n^{-1/2})$ (Gretton et al., 2012, Chapter 4.1) and $O(n^{-1/d})$ (Peyré & Cuturi, 2020, Chapter 8.4.1) suggests, that the exponent of $\mathcal{D}_K$ in Theorem 2 cannot be improved over $1/d$.

## 3 GRADIENTS OF SLICED MMD

Next, we consider the functional $\mathcal{F}_\nu \colon \mathcal{P}_2(\mathbb{R}^d) \to \mathbb{R}$ given by

$$\mathcal{F}_\nu(\mu) \coloneqq \mathcal{E}_K(\mu) + \mathcal{V}_{K,\nu}(\mu) = \mathcal{D}^2_K(\mu,\nu) + \mathrm{const}_\nu, \tag{2}$$

where $\mathcal{V}_{K,\nu}(\mu)$ is the so-called *potential energy*

$$\mathcal{V}_{K,\nu}(\mu) \coloneqq -\int_{\mathbb{R}^d} \int_{\mathbb{R}^d} K(x,y) \, \mathrm{d}\nu(y) \, \mathrm{d}\mu(x)$$

acting as an attraction term between the masses of $\mu$ and $\nu$, while the interaction energy $\mathcal{E}_K$ is a repulsion term enforcing a proper spread of $\mu$. For the rest of the paper, we always consider the negative distance kernel $K(x,y) \coloneqq -\|x-y\|$, which is the Riesz kernel (1) with $r = 1$. Then, we obtain directly from the metric property of MMD that the minimizer of the non-convex functional $\mathcal{F}_\nu$ is given by $\nu$. We are interested in computing gradient flows of $\mathcal{F}_\nu$ towards this minimizer. However, the computation of gradients in measure spaces for discrepancy functionals with non-smooth kernels is highly non-trivial and computationally costly, see e.g., (Altekrüger et al., 2023; Carrillo et al., 2020; Hertrich et al., 2023b).

As a remedy, we focus on a discrete form of the $d$-dimensional MMD. More precisely, we assume that $\mu$ and $\nu$ are empirical measures, i.e., they are of the form $\mu = \frac{1}{N}\sum_{i=1}^{N}\delta_{x_i}$ and $\nu = \frac{1}{M}\sum_{j=1}^{M}\delta_{y_j}$ for some $x_j, y_j \in \mathbb{R}^d$. Let $\boldsymbol{x} := (x_1, \ldots, x_N)$ and $\boldsymbol{y} := (y_1, \ldots, y_M)$. Then the functional $\mathcal{F}_\nu$ reduces to the function $F_d(\cdot|\boldsymbol{y})\colon \mathbb{R}^{Nd} \to \mathbb{R}$ given by

$$F_d(\boldsymbol{x}|\boldsymbol{y}) = -\frac{1}{2N^2}\sum_{i=1}^{N}\sum_{j=1}^{N}\|x_i - x_j\| + \frac{1}{MN}\sum_{i=1}^{N}\sum_{j=1}^{M}\|x_i - y_j\| \tag{3}$$

$$= \mathcal{D}_K^2\Big(\frac{1}{N}\sum_{i=1}^{N}\delta_{x_i}, \frac{1}{M}\sum_{j=1}^{M}\delta_{y_j}\Big) + \mathrm{const}_{\boldsymbol{y}}.$$

In order to evaluate the gradient of $F_d$ with respect to the support points $\boldsymbol{x}$, we use Theorem 1 to rewrite $F_d(\boldsymbol{x}|\boldsymbol{y})$ as

$$F_d(\boldsymbol{x}|\boldsymbol{y}) = c_d\mathbb{E}_{\xi\sim\mathcal{U}_{\mathbb{S}^{d-1}}}[F_1(\langle\xi, x_1\rangle, ..., \langle\xi, x_N\rangle|\langle\xi, y_1\rangle, ..., \langle\xi, y_M\rangle)].$$

Then, the gradient of $F_d$ with respect to $x_i$ is given by

$$\nabla_{x_i}F_d(\boldsymbol{x}|\boldsymbol{y}) = c_d\mathbb{E}_{\xi\sim\mathcal{U}_{\mathbb{S}^{d-1}}}[\partial_i F_1(\langle\xi, x_1\rangle, ..., \langle\xi, x_N\rangle|\langle\xi, y_1\rangle, ..., \langle\xi, y_M\rangle)\xi], \tag{4}$$

where $\partial_i F_1$ denotes the derivative of $F_1$ with respect to the $i$-th component of the input. Consequently, it suffices to compute gradients of $F_1$ in order to evaluate the gradient of $F_d$.

**A Sorting Algorithm for the 1D-Case.** Next, we derive a sorting algorithm to compute the gradient of $F_1$ efficiently. In particular, the proposed algorithm has complexity $O((M+N)\log(M+N))$ even though the definition of $F_1$ in (3) involves $N^2 + MN$ summands.

To this end, we split the functional $F_1$ into interaction and potential energy, i.e., $F_1(\boldsymbol{x}|\boldsymbol{y}) = E(\boldsymbol{x}) + V(\boldsymbol{x}|\boldsymbol{y})$ with

$$E(\boldsymbol{x}) := -\frac{1}{2N^2}\sum_{i=1}^{N}\sum_{j=1}^{N}|x_i - x_j|, \quad V(\boldsymbol{x}|\boldsymbol{y}) := \frac{1}{NM}\sum_{i=1}^{N}\sum_{j=1}^{M}|x_i - y_j|. \tag{5}$$

Then, we can compute the derivatives of $E$ and $V$ by the following theorem which proof is given in Appendix C.

**Theorem 3** (Derivatives of Interaction and Potential Energy)**.** *Let $x_1, ..., x_N \in \mathbb{R}$ be pairwise disjoint and $y_1, ..., y_M \in \mathbb{R}$ such that $x_i \neq y_j$ for all $i = 1, ..., N$ and $j = 1, ..., M$. Then, $E$ and $V$ are differentiable with*

$$\nabla_{x_i}E(\boldsymbol{x}) = \frac{N + 1 - 2\sigma^{-1}(i)}{N^2}, \quad \nabla_{x_i}V(\boldsymbol{x}|\boldsymbol{y}) = \frac{2\,\#\{j \in \{1, ..., M\} : y_j < x_i\} - M}{MN},$$

*where $\sigma\colon \{1, ..., N\} \to \{1, ..., N\}$ is the permutation with $x_{\sigma(1)} < ... < x_{\sigma(N)}$.*

Since $V$ is convex, we can show with the same proof that

$$\frac{2\,\#\{j \in \{1, ..., M\} : y_j < x_i\} - M}{MN} \in \partial_{x_i}V(\boldsymbol{x}|\boldsymbol{y}),$$

where $\partial_{x_i}$ is the subdifferential ov $V$ with respect to $x_i$ whenever $x_i = y_j$ for some $i, j$. By Theorem 3, we obtain that $\nabla F_1(\boldsymbol{x}|\boldsymbol{y}) = \nabla E(\boldsymbol{x}) + \nabla V(\boldsymbol{x}|\boldsymbol{v})$ can be computed by Algorithm 1 and Algorithm 2 with complexity $O(N\log(N))$ and $O((M+N)\log(M+N))$, respectively. The complexity is dominated by the sorting procedure. Both algorithms can be implemented in a vectorized form for computational efficiency. Note that by Lemma 9 from the appendix, the discrepancy with Riesz kernel and $r = 1$ can be represented by the cumulative distribution functions (cdfs) of the involved measures. Since the cdf of an one-dimensional empirical measure can be computed via sorting, we also obtain an $O((N+M)\log(M+N))$ algorithm for computing the one-dimensional MMD itself and not only for its derivative.

**Stochastic Approximation of Sliced MMD Gradients for $r = 1$.** To evaluate the gradient of $F_d$ efficiently, we use a stochastic gradient estimator. For $x_1, ..., x_N, y_1, ..., y_M \in \mathbb{R}^d$, we define for $P \in \mathbb{N}$ the stochastic gradient estimator of (4) as the random variable

$$\tilde{\nabla}_P F_d(\boldsymbol{x}|\boldsymbol{y}) = \Big(\tilde{\nabla}_{P,x_i}F_d(\boldsymbol{x}|\boldsymbol{y})\Big)_{i=1}^{N} \tag{6}$$

---

**Algorithm 1** Derivative of the interaction energy $E$ from (5).

---

**Input:** $x_1, ..., x_N \in \mathbb{R}$ with $x_i \neq x_j$ for $i \neq j$.
**Algorithm:**
Compute $\sigma_1, ...\sigma_N = \mathrm{argsort}(x_1, ..., x_N)$.
Compute $g_i = -\frac{2\sigma_i^{-1} - 1 - N}{N^2}$.
**Output:** $(g_1, ..., g_N) = \nabla E(x_1, ..., x_N)$.

---

---

**Algorithm 2** Derivative of the potential energy $V$ from (5).

---

**Input:** $x_1, ..., x_N \in \mathbb{R}$, $y_1, ..., y_M \in \mathbb{R}$ with $x_i \neq y_j$.
**Algorithm:**
Compute $\sigma_1, ..., \sigma_{N+M} = \mathrm{argsort}(x_1, ..., x_N, y_1, ..., y_M)$
Initialize $\tilde{h}_1 = \cdots = \tilde{h}_{M+N} = 0$.
**for** $j = 1, ..., M$ **do**
  Set $\tilde{h}_{\sigma(N+j)} = 1$.
**end for**
Set $h = 2\,\mathrm{cumsum}(\tilde{h}) - 1$
**for** $i = 1, ..., N$ **do**
  Set $g_i = \frac{h_{\sigma^{-1}(i)}}{MN}$,
**end for**
**Output:** $(g_1, ..., g_N) = \nabla V(x_1, ..., x_N | y_1, ..., y_M)$.

---

where

$$\tilde{\nabla}_{P,x_i} F_d(\boldsymbol{x}|\boldsymbol{y}) := \frac{c_d}{P} \sum_{p=1}^{P} \partial_i F_1(\langle \xi_p, x_1 \rangle, ..., \langle \xi_p, x_N \rangle | \langle \xi_p, y_1 \rangle, ..., \langle \xi_p, y_M \rangle) \xi_p,$$

for independent random variables $\xi_1, ..., \xi_P \sim \mathcal{U}_{\mathbb{S}^{d-1}}$. We obtain by (4) that $\tilde{\nabla} F_d$ is unbiased, i.e., $\mathbb{E}[\tilde{\nabla}_P F_d(\boldsymbol{x}|\boldsymbol{y})] = \nabla F_d(\boldsymbol{x}|\boldsymbol{y})$. Moreover, the following theorem shows that the error of $\tilde{\nabla}_P F_d$ converges to zero for a growing number $P$ of projections. The proof uses classical concentration inequalities and follows directly from Corollary 12 in Appendix D.

**Theorem 4** (Error Bound for Stochastic MMD Gradients)**.** *Let* $x_1, ..., x_N, y_1, ..., y_M \in \mathbb{R}^d$*. Then, it holds*

$$\mathbb{E}[\|\tilde{\nabla}_P F_d(\boldsymbol{x}|\boldsymbol{y}) - \nabla F_d(\boldsymbol{x}|\boldsymbol{y})\|] \in O\Big(\sqrt{\frac{d}{P}}\Big).$$

To verify this convergence rate numerically, we draw $N = 1000$ samples $x_1, ..., x_N$ from a Gaussian mixture model with two components and $M = 1000$ samples $y_1, ..., y_M$ from a Gaussian mixture model with ten components. The means are chosen randomly following a uniform distribution in $[-1, 1]^d$ and the standard deviation is set to $0.01$. Then, we compute numerically the expected relative approximation error between $\tilde{\nabla}_P F_d$ and $\nabla F_d$ for different choices of $P$ and $d$. The results are illustrated in the middle and in the right plot of Figure 1. We observe that this numerical evaluation underlines the convergence rate of $O\Big(\sqrt{\frac{d}{P}}\Big)$. In particular, the error scales with $O(\sqrt{d/P})$, which makes the approach applicable for high-dimensional problems.

**Remark 5** (Computational Complexity of Gradient Evaluations)**.** *The appearance of the $\sqrt{d}$ in the error bound is due to the scale factor $c_d$ between the MMD and the sliced MMD, which can be seen in the proof of Theorem 4. In particular, we require $O(d)$ projections in order to approximate $\nabla F_d(\boldsymbol{x}|\boldsymbol{y})$ by $\tilde{\nabla}_P F_d(\boldsymbol{x}|\boldsymbol{y})$ up to a fixed expected error of $\epsilon$. Together with the computational complexity of $O(dP(N + M)\log(N + M))$ for $\tilde{\nabla}_P F_d(\boldsymbol{x}, \boldsymbol{y})$, we obtain an overall complexity of $O(d^2(N + M)\log(N + M))$ in order to approximate $\nabla F_d(\boldsymbol{x}|\boldsymbol{y})$ up to an expected error of $\epsilon$. On the other hand, the naive computation of (gradients of) $F_d(\boldsymbol{x}|\boldsymbol{y})$ has a complexity of $O(d(N^2 + MN))$.*

*Consequently, we improve the quadratic complexity in the number of samples to $O(N \log(N))$. Here, we pay the price of quadratic instead of linear complexity in the dimension.*

## 4 GENERATIVE MMD FLOWS

In this section, we use MMD flows with the negative distance kernel for generative modelling. Throughout this section, we assume that we are given independent samples $y_1, ..., y_M \in \mathbb{R}^d$ from a target measure $\nu \in \mathcal{P}_2(\mathbb{R}^d)$ and define the empirical version of $\nu$ by $\nu_M \coloneqq \frac{1}{M} \sum_{i=1}^{M} \delta_{y_i}$.

### 4.1 MMD PARTICLE FLOWS

In order to derive a generative model approximating $\nu$, we simulate a gradient flow of the functional $\mathcal{F}_\nu$ from (2). Unfortunately, the computation of gradient flows in measure spaces for $\mathcal{F}_\nu$ is highly non-trivial and computationally costly, see (Altekrüger et al., 2023; Hertrich et al., 2023b). Therefore, we consider the (rescaled) gradient flow with respect to the functional $F_d$ instead. More precisely, we simulate for $F_d$ from (3), the (Euclidean) differential equation

$$\dot{\boldsymbol{x}} = -N \, \nabla F_d(\boldsymbol{x}|\boldsymbol{y}), \quad x(0) = (x_1^{(0)}, ..., x_N^{(0)}), \tag{7}$$

where the initial points $x_i^{(0)}$ are drawn independently from some measure $\mu_0 \in \mathcal{P}_2(\mathbb{R}^d)$. In our numerical experiments, we set $\mu_0$ to the uniform distribution on $[0, 1]^d$. Then, for any solution $x(t) = (x_1(t), ..., x_N(t))$ of (7), it is proven in (Altekrüger et al., 2023, Proposition 14) that the curve $\gamma_{M,N} \colon (0, \infty) \to \mathcal{P}_2(\mathbb{R}^d)$ defined by $\gamma_{M,N}(t) = \frac{1}{N} \sum_{i=1}^{N} \delta_{x_i(t)}$ is a Wasserstein gradient flow with respect to the function

$$\mathcal{F} \colon \mathcal{P}_2(\mathbb{R}^d) \to \mathbb{R} \cup \{\infty\}, \quad \mu \mapsto \begin{cases} \mathcal{F}_{\nu_M}, & \text{if } \mu = \frac{1}{N} \sum_{i=1}^{N} \delta_{x_i} \text{ for some } x_i \neq x_j \in \mathbb{R}^d, \\ +\infty, & \text{otherwise.} \end{cases}$$

Hence, we can expect for $M, N \to \infty$, that the curve $\gamma_{M,N}$ approximates the Wasserstein gradient flow with respect to $\mathcal{F}_\nu$. Consequently, we can derive a generative model by simulating the gradient flow (7). To this end, we use the explicit Euler scheme

$$\boldsymbol{x}^{(k+1)} = \boldsymbol{x}^{(k)} - \tau N \nabla F_d(\boldsymbol{x}^{(k)}|\boldsymbol{y}), \tag{8}$$

where $\boldsymbol{x}^{(k)} = (x_1^{(k)}, ..., x_N^{(k)})$ and $\tau > 0$ is some step size. Here, the gradient on the right-hand side can be evaluated very efficiently by the stochastic gradient estimator from (6).

**Momentum MMD Flows.** To reduce the required number of steps in (8), we introduce a momentum parameter. More precisely, for some given momentum parameter $m \in [0, 1)$ we consider the momentum MMD flow defined by the following iteration

$$\begin{aligned} \boldsymbol{v}^{(k+1)} &= \nabla F_d(\boldsymbol{x}^{(k)}|\boldsymbol{y}) + m \, \boldsymbol{v}^{(k)} \\ \boldsymbol{x}^{(k+1)} &= \boldsymbol{x}^{(k)} - \tau N \, \boldsymbol{v}^{(k+1)}, \end{aligned} \tag{9}$$

where $\tau > 0$ is some step size, $x_i^{(0)}$ are independent samples from a initial measure $\mu_0$ and $v_i^{(0)} = 0$. Note that the MMD flow (8) is a special case of the momentum MMD flow (9) with $m = 0$.

In Figure 2, we illustrate the momentum MMD flow (9) and MMD flow (8) without momentum from a uniform distribution on $[0, 1]^d$ to MNIST (LeCun et al., 1998) and CIFAR10 (Krizhevsky, 2009). The momentum is set to $m = 0.9$ for MNIST and to $m = 0.6$ for CIFAR10. We observe that the momentum MMD flow (9) converges indeed faster than the MMD flow (8) without momentum.

### 4.2 GENERATIVE MMD FLOWS

The (momentum) MMD flows from (8) and (9) transform samples from the initial distribution $\mu_0$ into samples from the target distribution $\nu$. Therefore, we propose to train a generative model which approximates these schemes. The main idea is to approximate the Wasserstein gradient flow $\gamma \colon [0, \infty) \to \mathcal{P}_2(\mathbb{R}^d)$ with respect to $\mathcal{F}_\nu$ from (2) starting at some latent distribution $\mu_0 = \gamma(0)$. Then, we iteratively train neural networks $\Phi_1, ..., \Phi_L$ such that $\gamma(t_l) \approx \Phi_{l\#}\gamma(t_{l-1})$ for some $t_l$ with

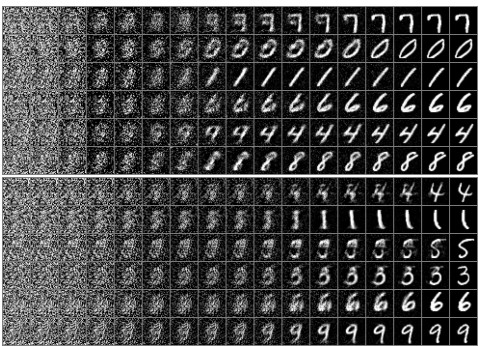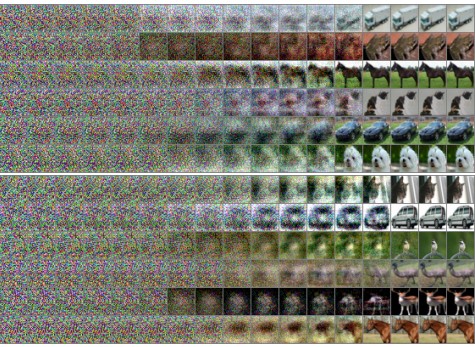

Figure 2: Samples and their trajectories from MNIST (left) and CIFAR10 (right) in the MMD flow with momentum (9, top) and without momentum (8, bottom) starting in the uniform distribution on $[0,1]^d$ after $2^k$ steps with $k \in \{0, ..., 16\}$ (for MNIST) and $k \in \{3, ..., 19\}$ (for CIFAR10). We observe that the momentum MMD flow (9) converges faster than the MMD flow (8) without momentum.

$0 = t_0 < t_1 < \cdots < t_L$. Then, for $t_L$ large enough, it holds $\nu \approx \gamma(t_L) \approx (\Phi_L \circ \cdots \circ \Phi_1)_\# \gamma(0)$ with $\gamma(0) = \mu_0$. Such methods learning iteratively an "interpolation path" are exploited several times in literature, e.g., Arbel et al. (2021); Fan et al. (2022); Ho et al. (2020). To implement this numerically, we train each network $\Phi_l$ such that it approximates $T_l$ number of steps from (8) or (9). The training procedure of our generative MMD flow is summarized in Algorithm 3 in Appendix F. Once the networks $\Phi_1, ..., \Phi_L$ are trained, we can infer a new sample $x$ from our (approximated) target distribution $\nu$ as follows. We draw a sample $x^{(0)}$ from $\mu_0$, compute $x^{(l)} = x^{(l-1)} - \Phi_l(x^{(l-1)})$ for $l = 1, ..., L$ and set $x = x^{(L)}$. In particular, this allows us to simulate paths of the discrepancy flow we have not trained on.

**Remark 6** (Iterative Training and Sampling). *Since the networks are not trained in an end-to-end fashion but separately, their GPU memory load is relatively low despite a high number of trainable parameters of the full model $(\Phi_l)_{l=1}^L$. This enables training of our model on an 8 GB GPU. Moreover, the training can easily be continued by adding additional networks $\Phi_l, l = L+1, ..., L'$ to an already trained generative MMD flow $(\Phi_l)_{l=1}^L$, which makes applications more flexible.*

## 5 NUMERICAL EXAMPLES

In this section, we apply generative MMD flows for image generation on MNIST, FashionMNIST (Xiao et al., 2017),CIFAR10 and CelebA (Liu et al., 2015). The images from MNIST and Fashion-MNIST are $28 \times 28$ gray-value images, while CIFAR10 consists of $32 \times 32$ RGB images resulting in the dimensions $d = 784$ and $d = 3072$, respectively. For CelebA, we centercrop the images to $140 \times 140$ and then bicubicely resize them to $64 \times 64$. We run all experiments either on a single NVIDIA GeForce RTX 3060 or a RTX 4090 GPU with 12GB or 24GB memory,respectively. To evaluate our results, we use the Fréchet inception distance (FID) (Heusel et al., 2017)[1] between

---

[1] We use the implementation from `https://github.com/mseitzer/pytorch-fid`.

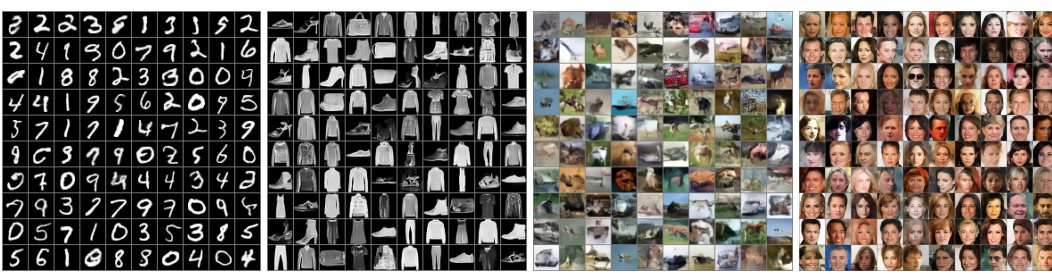

Figure 3: Generated samples of our generative MMD Flow.

Table 1: FID scores for different datasets and various methods.

| Method | MNIST | FashionMNIST | CIFAR10 | CelebA |
|---|---|---|---|---|
| *Auto-encoder based* | | | | |
| CWAE (Knop et al., 2020) | 23.6 | 50.0 | 120.0 | 49.7 |
| SWF+ Autoencoder + RealNVP (Bonet et al., 2022b) | 17.8 | 40.6 | - | 90.9 |
| 2-stage VAE (Dai & Wipf, 2019) | 12.6 | 29.3 | 72.9 | 44.4 |
| GLF (Xiao et al., 2019) | 8.2 | 21.3 | 88.3 | 53.2 |
| *Adversarial* | | | | |
| WGAN (Arjovsky et al., 2017; Lucic et al., 2018) | 6.7 | 21.5 | 55.2 | 41.3 |
| MMD GAN (Bińkowski et al., 2018) | 4.2 | - | 48.1 | **29.2** |
| *Score-based* | | | | |
| NCSN (Song & Ermon, 2019) | - | - | **25.3** | - |
| *Flow based* | | | | |
| SWF (Liutkus et al., 2019) [3] | 225.1 | 207.6 | - | 91.2 |
| SIG (Dai & Seljak, 2021) | 4.5 | 13.7 | 66.5 | 37.3 |
| $\ell$-SWF (Du et al., 2023) | - | - | 59.7 | 38.3 |
| Generative Sliced MMD Flow (ours) | $\mathbf{3.1} \pm 0.06$ | $\mathbf{11.3}\pm 0.07$ | $54.8 \pm 0.26$ | $32.1\pm 0.17$ |

10K generated samples and the test dataset. Here, a smaller FID value indicates a higher similarity between generated and test samples.

We choose the networks $(\Phi_l)_{l=1}^{L}$ to be UNets (Ronneberger et al., 2015), where we use the implementation from (Huang et al., 2021) based on (Ho et al., 2020). Then, we run the generative MMD flow for $L = 55$ (MNIST), $L = 67$ (FashionMNIST), $L = 86$ (CIFAR10) and $L = 71$ (CelebA) networks. The exact setup is described in Appendix H. We compare the resulting FIDs with other gradient-flow-based models and various further methods in Table 1. We computed the standard deviations by independently sampling ten times from one training run and computing the corresponding FID. We observe that we achieve excellent performance on MNIST and FashionMNIST as well as very good results on CIFAR10 and CelebA. Generated samples are given in Figure 3 and more samples are given in Appendix I. The $L2$-nearest neighbors of the generated samples on MNIST are also illustrated in Figure 8 in Appendix I.

## 6 Conclusions

**Discussion.** We introduced an algorithm to compute (gradients) of the MMD with Riesz kernel efficiently via slicing and sorting reducing the dependence of the computational complexity on the number of particles from $O(NM + N^2)$ to $O((N + M) \log(N + M))$. For the implementations, we approximated the gradient of sliced MMD by a finite number of slices and proved that the corresponding approximation error depends by a square root on the dimension. We applied our algorithm for computing MMD flows and approximated them by neural networks. Here, a sequential learning approach ensures computational efficiency. We included numerical examples for image generation on MNIST, FashionMNIST and CIFAR10.

**Limitations.** One of the disadvantages of interacting particle methods is that batching is not easily possible: The particle flow for one set of training points does not give a helpful approximation for another set of training points. This is due to the interaction energy and a general problem of particle flows. Furthermore, taking the projections involves multiplication of every data point with a "full" projection and therefore scales with the dimension $d$. Taking "local" projections like in (Du et al., 2023; Nguyen & Ho, 2022) can be much more efficient.

**Outlook.** Our paper is the first work which utilizes sliced MMD flows for generative modelling. Consequently the approach can be extended in several directions. Other kernels are considered in the context of slicing in the follow-up paper (Hertrich, 2024). From a theoretical viewpoint, the derivative formulas from Theorem 3 can be extended to the non-discrete case by the use of quantile functions, see (Bonaschi et al., 2015; Hertrich et al., 2023a) for some first approaches into this direction. Towards applications, we could extend the framework to posterior sampling in Bayesian inverse problems. In this context, the fast computation of MMD gradients can be also of interest for applications which are not based on gradient flows, see e.g., Ardizzone et al. (2019). Finally, the consideration of sliced probability metrics is closely related to the Radon transform and is therefore of interest also for non-Euclidean domains like the sphere, see e.g., (Bonet et al., 2022a; Quellmalz et al., 2023).

---

[3] values taken from Bonet et al. (2022b)

ACKNOWLEDGEMENTS

Many thanks to J. Chemseddine for providing parts of the proof of Theorem 2, and to R. Beinert and G. Steidl for fruitful discussions. We thank Mitchell Krock for finding a typo. J.H. acknowledges funding by the German Research Foundation (DFG) within the project STE 571/16-1 and by the EPSRC programme grant "The Mathematics of Deep Learning" with reference EP/V026259/1, C.W. by the DFG within the SFB "Tomography Across the Scales" (STE 571/19-1, project number: 495365311), F.A. by the DFG under Germany's Excellence Strategy – The Berlin Mathematics Research Center MATH+ (project AA 5-6), and P.H. from the DFG within the project SPP 2298 "Theoretical Foundations of Deep Learning".

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

# A  PROOF OF THEOREM 1

Let $\mathcal{U}_{\mathbb{S}^{d-1}}$ be the uniform distribution on $\mathbb{S}^{d-1}$ and let $\mathrm{k}(x,y) = -|x-y|^r$ for $x, y \in \mathbb{R}, x \neq y$ and $r \in (0,2)$. Moreover, denote by $e = (1, ..., 0) \in \mathbb{S}^{d-1}$ the first unit vector. Then, we have for $x, y \in \mathbb{R}^d$ that

$$K(x,y) := -\int_{\mathbb{S}^{d-1}} |\langle \xi, x\rangle - \langle \xi, y\rangle|^r \mathrm{d}\mathcal{U}_{\mathbb{S}^{d-1}}(\xi) = -\|x-y\|^r \int_{\mathbb{S}^{d-1}} \left| \left\langle \xi, \frac{x-y}{\|x-y\|} \right\rangle \right|^r \mathrm{d}\mathcal{U}_{\mathbb{S}^{d-1}}(\xi)$$

$$= -\|x-y\|^r \int_{\mathbb{S}^{d-1}} |\langle \xi, e\rangle|^r \mathrm{d}\mathcal{U}_{\mathbb{S}^{d-1}}(\xi) = -\|x-y\|^r \underbrace{\int_{\mathbb{S}^{d-1}} |\xi_1|^r \mathrm{d}\mathcal{U}_{\mathbb{S}^{d-1}}(\xi)}_{=:c_{d,r}^{-1}}.$$

It remains to compute the constant $c_{d,r}$ which is straightforward for $d = 1$. For $d > 1$ the map

$$(t, \eta) \mapsto (t, \eta\sqrt{1-t^2})$$

defined on $[-1, 1) \times \mathbb{S}^{d-2}$ is a parametrization of $\mathbb{S}^{d-1}$. The surface measure on $\mathbb{S}^{d-1}$ is then given by

$$\mathrm{d}\sigma_{\mathbb{S}^{d-1}}(\xi) = (1-t^2)^{\frac{d-3}{2}} \mathrm{d}\sigma_{\mathbb{S}^{d-2}}(\eta)\mathrm{d}t,$$

see (Atkinson & Han, 2012, Eq. 1.16). Furthermore, the uniform surface measure $\mathcal{U}_{\mathbb{S}^{d-1}}$ reads as

$$\mathrm{d}\mathcal{U}_{\mathbb{S}^{d-1}}(\xi) = \frac{1}{s_{d-1}}(1-t^2)^{\frac{d-3}{2}} \mathrm{d}\sigma_{\mathbb{S}^{d-2}}(\eta)\mathrm{d}t,$$

where $s_{d-1}$ is the volume of $\mathbb{S}^{d-1}$. Hence

$$c_{d,r}^{-1} = \int_{\mathbb{S}^{d-1}} |\xi_1|^r \mathrm{d}\mathcal{U}_{\mathbb{S}^{d-1}}(\xi) = \frac{1}{s_{d-1}} \int_{\mathbb{S}^{d-2}} \int_{-1}^{1} |t|^r (1-t^2)^{\frac{d-3}{2}} \mathrm{d}t \mathrm{d}\sigma_{\mathbb{S}^{d-2}}(\eta) \qquad (10)$$

$$= \frac{s_{d-2}}{s_{d-1}} 2 \int_0^1 t^r (1-t^2)^{\frac{d-3}{2}} \mathrm{d}t = \frac{s_{d-2}}{s_{d-1}} B\left(\frac{r+1}{2}, \frac{d-1}{2}\right),$$

where $B(z_1, z_2)$ is the beta function and we used the integral identity

$$B(z_1, z_2) = 2\int_0^1 t^{2z_1 - 1}(1-t^2)^{z_2-1}\mathrm{d}t.$$

Finally, noting that $s_{d-1} = \frac{2\pi^{d/2}}{\Gamma(\frac{d}{2})}$ and $\mathrm{B}(z_1, z_2) = \frac{\Gamma(z_1)\Gamma(z_2)}{\Gamma(z_1+z_2)}$, (10) can be computed as

$$c_{d,r}^{-1} = \frac{\Gamma(\frac{d}{2})}{\sqrt{\pi}\Gamma(\frac{d-1}{2})} \frac{\Gamma(\frac{r+1}{2})\Gamma(\frac{d-1}{2})}{\Gamma(\frac{r+d}{2})} = \frac{\Gamma(\frac{d}{2})\Gamma(\frac{r+1}{2})}{\sqrt{\pi}\Gamma(\frac{d+r}{2})}$$

Taking the inverse gives the claim. $\qquad \square$

**Remark 7** (Extension to $\mathcal{P}_{\frac{r}{2}}(\mathbb{R}^d)$). *We can extend Theorem 1 to $\mathcal{P}_{\frac{r}{2}}(\mathbb{R}^d)$. To this end, we first show, how we can deduce from (Modeste & Dombry, 2023, Prop 2.14) that the MMD $\mathcal{D}_{\tilde{K}}$ with respect to the extended Riesz kernel $\tilde{K}(x,y) = -\|x-y\|^r + \|x\|^r + \|y\|^r$ defines a metric on $\mathcal{P}_{\frac{r}{2}}(\mathbb{R}^d)$. Second, we will see that Theorem 1 can be extended to $\mathcal{P}_{\frac{r}{2}}(\mathbb{R}^d)$ as well.*

*(i) By (Steinwart & Christmann, 2008, Thm 4.26) the MMD*

$$\mathcal{D}_{\tilde{K}}^2(\mu, \nu) = \int \int \tilde{K}(x,y)d(\mu-\nu)(x)d(\mu-\nu)(y)$$

*is finite on $\mathcal{M}_{\tilde{K}} = \{\mu \in \mathcal{M}(\mathbb{R}^d) : \int \sqrt{\tilde{K}(x,x)}\mathrm{d}|\mu| < \infty\}$, where $\mathcal{M}(\mathbb{R}^d)$ is the space of all signed measures on $\mathbb{R}^d$. Since $\tilde{K}(x,x) = 2\|x\|^r$, we have that $\mathcal{M}_{\tilde{K}} = \{\mu \in \mathcal{M}(\mathbb{R}^d) : \int \|x\|^{\frac{r}{2}}\mathrm{d}|\mu| < \infty\}$ such that $\mathcal{M}_{\tilde{K}} \cap \mathcal{P}(\mathbb{R}^d) = \mathcal{P}_{\frac{r}{2}}(\mathbb{R}^d)$. Now, inserting $\tilde{K}$ (Modeste & Dombry, 2023, Prop 2.14) states that $\mathcal{D}_{\tilde{K}}$ is a metric on $\mathcal{M}_{\tilde{K}} \cap \mathcal{P}(\mathbb{R}^d) = \mathcal{P}_{\frac{r}{2}}(\mathbb{R}^d)$, where the assumptions of the proposition are checked in (Modeste & Dombry, 2023, Ex 2, Ex 3) and $\tilde{K}$ is named $k_H$ with $H = \frac{r}{2}$.*

*(ii) Let $\tilde{K} = c_{d,r}^{-1}\tilde{K}$ be a rescaled and $\tilde{k}(x,y) = -|x-y|^r + |x|^r + |y|^r$ be the one-dimensional version of the extended Riesz kernel $\tilde{K}$. With the same calculations as in the proof of Theorem 1 we can show that*

$$\tilde{K}(x,y) = \int_{\mathbb{S}^{d-1}} \tilde{k}(\langle \xi, x \rangle, \langle \xi, y \rangle) d\mathcal{U}_{\mathbb{S}^{d-1}}(\xi).$$

*Thus, we also have that $\mathcal{SD}_{\tilde{k}} = \mathcal{D}_{\tilde{K}}$ on $\mathcal{P}_{\frac{r}{2}}(\mathbb{R}^d)$.*

## B    PROOF OF THEOREM 2

In the first part of the proof, we will use properties of reproducing kernel Hilbert spaces (RKHS), see (Steinwart & Christmann, 2008) for an overview on RKHS. For the second part, we will need two lemmata. The first one gives a definition of the MMD in terms of the Fourier transform (characteristic function) of the involved measures, where $\hat{\mu}(\xi) = \langle e^{-ix\xi}, \mu \rangle = \int_{\mathbb{R}^d} e^{-i\langle \xi, x \rangle} d\mu(x)$. Its proof can be found, e.g., in (Székely & Rizzo, 2013, Proposition 2). Note that the constant on the right hand-side differs from (Székely & Rizzo, 2013, Proposition 2), since we use a different notion of the Fourier transform and the constant $\frac{1}{2}$ in the MMD.

**Lemma 8.** *Let $\mu \in \mathcal{P}_1(\mathbb{R})$ and $K(x,y) = -\|x-y\|$. Then its holds*

$$\mathcal{D}_K^2(\mu,\nu) = \frac{\Gamma(\frac{d+1}{2})}{2\pi^{\frac{d+1}{2}}} \int_{\mathbb{R}^d} \frac{|\hat{\mu}(\xi) - \hat{\nu}(\xi)|^2}{\|\xi\|^{1+d}} d\xi.$$

For the next lemma, recall that the cumulative density functions (cdf) of $\mu \in \mathcal{P}(\mathbb{R})$ is the function $F_\mu : \mathbb{R} \to [0,1]$ defined by

$$F_\mu(x) \coloneqq \mu((-\infty, x]) = \int_{\mathbb{R}} \chi_{(-\infty,x]}(y) d\mu(y), \quad \chi_A(x) = \begin{cases} 1, & \text{if } x \in A, \\ 0, & \text{otherwise.} \end{cases}$$

We will need that the *Cramer distance* between probability measures $\mu, \nu$ with cdfs $F_\mu$ and $F_\nu$ defined by

$$\ell_p(\mu,\nu) \coloneqq \left( \int_{\mathbb{R}} |F_\mu - F_\nu|^p dx \right)^{\frac{1}{p}},$$

if it exists. The Cramer distance does not exist for arbitrary probability measures. However, for $\mu, \nu \in \mathcal{P}_1(\mathbb{R})$ is well-known that

$$\ell_1(\mu,\nu) = \mathcal{W}_1(\mu,\nu) \tag{11}$$

and we have indeed $F_\mu - F_\nu \in L_2(\mathbb{R})$. The following relation can be found in the literature, see, e.g., Székely (2002). However, we prefer to add a proof which clearly shows which assumptions are necessary.

**Lemma 9.** *Let $\mu, \nu \in \mathcal{P}_1(\mathbb{R})$ and $k(x.y) = -|x-y|$. Then it holds*

$$\ell_2(\mu,\nu) = \mathcal{D}_k(\mu,\nu).$$

*Proof.* By Lemma 8, we know that

$$\mathcal{D}_k^2(\mu,\nu) = -\frac{1}{2} \int_{\mathbb{R}} \int_{\mathbb{R}} |x-y| d(\mu-\nu)(x) d(\mu-\nu)(y) = \frac{1}{2\pi} \int_{\mathbb{R}} \frac{|\hat{\mu}(\xi) - \hat{\nu}(\xi)|^2}{\xi^2} d\xi. \tag{12}$$

For $\mu, \nu \in \mathcal{P}_1(\mathbb{R})$, we have $F_\mu - F_\nu \in L_2(\mathbb{R})$ and can apply Parseval's equality

$$\ell_2^2(\mu,\nu) = \int_{\mathbb{R}} |F_\mu(t) - F_\nu(t)|^2 dt = \frac{1}{2\pi} \int_{\mathbb{R}} |\hat{F}_\mu(\xi) - \hat{F}_\nu(\xi)|^2 d\xi. \tag{13}$$

Now we have for the distributional derivative of $F_\mu$ that $DF_\mu = \mu$, which can be seen as follows: using Fubini's theorem, we have for any Schwartz function $\phi \in \mathcal{S}(\mathbb{R})$ that

$$\langle DF_\mu, \phi \rangle = -\langle F_\mu, \phi' \rangle = -\int_{\mathbb{R}} \int_{\mathbb{R}} \chi_{(-\infty,x]}(y) \phi'(x) d\mu(y) dx = -\int_{\mathbb{R}} \int_{\mathbb{R}} \chi_{(-\infty,x]}(y) \phi'(x) dx d\mu(y)$$

$$= -\int_{\mathbb{R}} \int_y^\infty \phi'(x) dx d\mu(y) = \int_{\mathbb{R}} \phi(y) d\mu(y) = \langle \mu, \phi \rangle.$$

Then we obtain by the differentiation property of the Fourier transform (Plonka et al., 2018) that

$$\hat{\mu}(\xi) = \widehat{DF_\mu}(\xi) = -i\xi \hat{F}_\mu(\xi).$$

Finally, (13) becomes

$$\ell_2^2(\mu, \nu) = \frac{1}{2\pi} \int_{\mathbb{R}} \frac{|\hat{\mu}(\xi) - \hat{\nu}(\xi)|^2}{\xi^2} \, \mathrm{d}\xi,$$

which yields the assertion by (12). $\qquad\square$

Now we can prove Theorem 2.

*Proof.* 1. To prove of the first inequality, we use the reproducing property

$$\langle f, \tilde{K}(x, \cdot) \rangle_{\mathcal{H}_{\tilde{K}}} = f(x)$$

of the kernel in the associated RKHS $\mathcal{H}_{\tilde{K}}$. For any $f \in \mathcal{H}_{\tilde{K}}$ with $\|f\|_{\mathcal{H}_{\tilde{K}}} \leq 1$ and any $\pi \in \Pi(\mu, \nu)$, we use the estimation

$$\left| \int_{\mathbb{R}^d} f(x) \mathrm{d}(\mu - \nu)(x) \right| = \left| \int_{\mathbb{R}^d} \int_{\mathbb{R}^d} f(x) - f(y) \mathrm{d}\pi(x, y) \right| \leq \int_{\mathbb{R}^d} \int_{\mathbb{R}^d} |f(x) - f(y)| \mathrm{d}\pi(x, y)$$

$$= \int_{\mathbb{R}^d} \int_{\mathbb{R}^d} |\langle f, \tilde{K}(x, \cdot) - \tilde{K}(y, \cdot) \rangle_{\mathcal{H}_{\tilde{K}}}| \mathrm{d}\pi(x, y)$$

$$\leq \int_{\mathbb{R}^d} \int_{\mathbb{R}^d} \|\tilde{K}(x, \cdot) - \tilde{K}(y, \cdot)\|_{\mathcal{H}_{\tilde{K}}} \mathrm{d}\pi(x, y),$$

which is called "coupling bound" in (Sriperumbudur et al., 2010, Prop. 20). Then, since $\|\tilde{K}(x, \cdot) - \tilde{K}(y, \cdot)\|_{\mathcal{H}_{\tilde{K}}}^2 = \tilde{K}(x, x) + \tilde{K}(y, y) - 2\tilde{K}(x, y) = 2\|x - y\|$ and using Jensen's inequality for the concave function $\sqrt{\cdot}$, we obtain

$$\left| \int_{\mathbb{R}^d} f(x) \mathrm{d}(\mu - \nu)(x) \right| \leq \sqrt{2} \int_{\mathbb{R}^d} \int_{\mathbb{R}^d} \|x - y\|^{\frac{1}{2}} \mathrm{d}\pi(x, y) \leq \left( 2 \int_{\mathbb{R}^d} \int_{\mathbb{R}^d} \|x - y\| \mathrm{d}\pi(x, y) \right)^{\frac{1}{2}}.$$

By the dual definition of the discrepancy $2\mathcal{D}_K(\mu, \nu) = \sup_{\|f\|_{\mathcal{H}_{\tilde{K}}} \leq 1} \int_{\mathbb{R}^d} f \mathrm{d}(\mu - \nu)$, see Novak & Wozniakowski (2010), and taking the supremum over all such $f$ and the infimum over all $\pi \in \Pi(\mu, \nu)$, we finally arrive at

$$2\mathcal{D}_K^2(\mu, \nu) \leq \mathcal{W}_1(\mu, \nu).$$

2. The second inequality can be seen as follows: by Bonnotte (2013, Lemma 5.1.4), there exists a constant $c_d > 0$ such that

$$W_1(\mu, \nu) \leq c_d R^{\frac{d}{d+1}} \mathcal{SW}_1(\mu, \nu)^{\frac{1}{d+1}} = c_d R^{\frac{d}{d+1}} \left( \mathbb{E}_{\xi \sim \mathcal{U}_{\mathbb{S}^{d-1}}} \left[ \mathcal{W}_1 \left( P_{\xi \#} \mu, P_{\xi \#} \nu \right) \right] \right)^{\frac{1}{d+1}}.$$

Further, we obtain by (11), the Cauchy-Schwarz inequality and Lemma 9 that

$$\mathbb{E}_{\xi \sim \mathcal{U}_{\mathbb{S}^{d-1}}} \left[ W_1 \left( P_{\xi \#} \mu, P_{\xi \#} \nu \right) \right] = \mathbb{E}_{\xi \sim \mathcal{U}_{\mathbb{S}^{d-1}}} \left[ l_1 \left( P_{\xi \#} \mu, P_{\xi \#} \nu \right) \right]$$

$$\leq \mathbb{E}_{\xi \sim \mathcal{U}_{\mathbb{S}^{d-1}}} \left[ (2R)^{\frac{1}{2}} l_2 \left( P_{\xi \#} \mu, P_{\xi \#} \nu \right) \right]$$

$$= (2R)^{\frac{1}{2}} \mathbb{E}_{\xi \sim \mathcal{U}_{\mathbb{S}^{d-1}}} \left[ \mathcal{D}_k \left( P_{\xi \#} \mu, P_{\xi \#} \nu \right) \right]$$

$$\leq (2R)^{\frac{1}{2}} \left( \mathbb{E}_{\xi \sim \mathcal{U}_{\mathbb{S}^{d-1}}} \left[ \mathcal{D}_k^2 \left( P_{\xi \#} \mu, P_{\xi \#} \nu \right) \right] \right)^{\frac{1}{2}},$$

and finally by Theorem 1 that

$$\mathbb{E}_{\xi \sim \mathcal{U}_{\mathbb{S}^{d-1}}} \left[ W_1 \left( P_{\xi \#} \mu, P_{\xi \#} \nu \right) \right] \leq (2R)^{\frac{1}{2}} \mathcal{D}_K(\mu, \nu) = (2R)^{\frac{1}{2}} \left( \frac{\Gamma(\frac{d}{2})}{\sqrt{\pi} \Gamma(\frac{d+1}{2})} \right)^{\frac{1}{2}} \mathcal{D}_K(\mu, \nu).$$

In summary, this results in

$$W_1(\mu, \nu) \leq c_d \left( \frac{2\Gamma(\frac{d}{2})}{\sqrt{\pi} \Gamma(\frac{d+1}{2})} \right)^{\frac{1}{2(d+1)}} R^{\frac{2d+1}{2d+2}} \mathcal{D}_K(\mu, \nu)^{\frac{1}{d+1}}.$$

$\qquad\square$

Under some additional assumptions, similar bounds have been considered in Chafaï et al. (2018) for the Coloumb kernel $K(x, y) = \|x - y\|^{2-d}$.

## C  PROOF OF THEOREM 3

**Interaction Energy:** This part of the proof is similar to (Teuber et al., 2011, Sec. 3). Using that $\sigma$ is a permutation and by reordering the terms in the double sum, we can rewrite the interaction energy by

$$E(\boldsymbol{x}) = -\frac{1}{2N^2}\sum_{i=1}^{N}\sum_{j=1}^{N}|x_i - x_j| = -\frac{1}{2N^2}\sum_{i=1}^{N}\sum_{j=1}^{N}|x_{\sigma(i)} - x_{\sigma(j)}|$$

$$= -\frac{1}{N^2}\sum_{i=1}^{N}\sum_{j=i+1}^{N} x_{\sigma(j)} - x_{\sigma(i)} = \sum_{i=1}^{N}\frac{N-(2i-1)}{N^2}x_{\sigma(i)} = \sum_{i=1}^{N}\frac{N-(2\sigma^{-1}(i)-1)}{N^2}x_i.$$

Since the $x_i$ are pairwise disjoint, the sorting permutation $\sigma$ is constant in a neighborhood of $\boldsymbol{x}$. Hence, $E$ is differentiable with derivative

$$\nabla_{x_i} E(\boldsymbol{x}) = \frac{N+1-2\sigma^{-1}(i)}{N^2}.$$

**Potential Energy:** For any $x \neq y \in \mathbb{R}$ it holds

$$\nabla_x |x - y| = \chi(x, y), \quad \text{where} \quad \chi(x, y) = \begin{cases} 1, & \text{if } x > y, \\ -1, & \text{if } x < y. \end{cases}$$

Thus, we have that

$$\nabla_{x_i} V(\boldsymbol{x}|\boldsymbol{y}) = \frac{1}{MN}\sum_{j=1}^{M}\chi(x_i, y_j)$$

$$= \frac{1}{MN}\big(\#\{j \in \{1, ..., M\} : y_j < x_i\} - \#\{j \in \{1, ..., M\} : y_j > x_i\}\big)$$

Using that $\#\{j \in \{1, ..., M\} : y_j > x_i\} = M - \#\{j \in \{1, ..., M\} : y_j < x_i\}$ the above expression is equal to

$$\frac{1}{MN}\big(2\,\#\{j \in \{1, ..., M\} : y_j < x_i\} - M\big) = \frac{2\,\#\{j \in \{1, ..., M\} : y_j < x_i\} - M}{MN}.$$

$\square$

## D  PROOF OF THEOREM 4

In this section, we derive error bounds for the stochastic estimators for the gradient of MMD as defined in (6) for the Riesz kernel with $r = 1$. To this end, we employ concentration inequalities (Vershynin, 2018), which were generalized to vector-valued random variables in Kohler & Lucchi (2017). We will need the following Lemma which is the Bernstein inequality and is stated in (Kohler & Lucchi, 2017, Lemma 18).

**Lemma 10** (Bernstein Inequality). *Let $X_1, ..., X_P$ be independent random vectors with mean zero, $\|X_i\| \leq \mu$ and $\mathbb{E}[\|X_i\|_2^2] \leq \sigma^2$. Then it holds for $0 < t < \frac{\sigma^2}{\mu}$ that*

$$\mathbb{P}\Big[\Big\|\frac{1}{P}\sum_{i=1}^{P}X_i\Big\|_2 > t\Big] \leq \exp\Big(-\frac{P\,t^2}{8\sigma^2} + \frac{1}{4}\Big).$$

Now we can use Lemma 10 to show convergence of the finite sum approximation to the exact gradient.

**Theorem 11** (Concentration Inequality). *Let $x_1, ...x_N, y_1, ..., y_M \in \mathbb{R}^d$. Then, it holds*

$$\mathbb{P}\Big[\|\tilde{\nabla}_P F_d(\boldsymbol{x}|\boldsymbol{y}) - \nabla F_d(\boldsymbol{x}|\boldsymbol{y})\| > t\Big] \leq \exp\Big(-\frac{P\,t^2}{32\,(c_d+1)^2} + \frac{1}{4}\Big).$$

*Proof.* Let $\xi_1, ..., \xi_P \sim \mathcal{U}_{\mathbb{S}^{d-1}}$ be the independent random variables from the definition of $\tilde{\nabla}_P$ in (6). We set

$$X_{i,l} := c_d \nabla_l F_1(\langle \xi_i, x_1 \rangle, ..., \langle \xi_i, x_N \rangle | \langle \xi_i, y_1 \rangle, ..., \langle \xi_i, y_M \rangle) \xi_i$$

and define the $dN$-dimensional random vector $X_i = (X_{i,1}, \cdots X_{i,N})$. Then, we have by (4) that

$$\mathbb{E}[X_i] = \nabla F_d(\boldsymbol{x}|\boldsymbol{y}) = \left( \nabla_{x_1} F_d(\boldsymbol{x}|\boldsymbol{y}), \cdots, \nabla_{x_N} F_d(\boldsymbol{x}|\boldsymbol{y}) \right).$$

By Theorem 3, we know that $\|X_{i,l}\|_2 \leq \frac{2c_d}{N}$, and by (3) it holds

$$\|\mathbb{E}[X_{i,l}]\|_2 = \|\nabla_{x_l} F_d(\boldsymbol{x}|\boldsymbol{y})\|_2 \leq \frac{2}{N}.$$

Let $\tilde{X}_i = X_i - \mathbb{E}[X_i]$. Then it holds

$$\|\tilde{X}_i\|_2 \leq \sum_{l=1}^{N} \|\tilde{X}_{i,l}\|_2 \leq \sum_{l=1}^{N} \frac{2c_d + 2}{N} = 2c_d + 2$$

and thus $\mathbb{E}[\|\tilde{X}_i\|_2^2] \leq 4(c_d + 1)^2$. Since we have $\mathbb{E}[\tilde{X}_i] = 0$ for all $i = 1, ..., P$, we can apply Lemma 10 and obtain

$$\mathbb{P}\left[ \left\| \frac{1}{P} \sum_{i=1}^{P} X_i - \nabla F_d(\boldsymbol{x}|\boldsymbol{y}) \right\| > t \right] = \mathbb{P}\left[ \left\| \frac{1}{P} \sum_{i=1}^{P} \tilde{X}_i \right\| > t \right] \leq \exp\left( -\frac{P\, t^2}{32\, (c_d + 1)^2} + \frac{1}{4} \right).$$

Since we have by definition that

$$\frac{1}{P} \sum_{i=1}^{P} X_i = \tilde{\nabla}_P F_d(\boldsymbol{x}|\boldsymbol{y})$$

this yields the assertion. $\square$

Finally we can draw a corollary which immediately shows Theorem 4.

**Corollary 12** (Error Bound for Stochastic Gradients). *For $x_1, ..., x_N, y_1, ..., y_M \in \mathbb{R}^d$, it holds*

$$\mathbb{E}[\|\tilde{\nabla}_P F_d(\boldsymbol{x}|\boldsymbol{y}) - \nabla F_d(\boldsymbol{x}|\boldsymbol{y})\|] \leq \frac{\exp(1/4)\sqrt{32}\pi(\sqrt{d} + 1)}{2\sqrt{2\, P}}.$$

*Proof.* Denote by $X$ the random variable

$$X = \|\tilde{\nabla}_P F_d(\boldsymbol{x}|\boldsymbol{x}) - \nabla F_d(\boldsymbol{x}|\boldsymbol{y})\|.$$

Then, we have by Theorem 11 that

$$\mathbb{P}[X > t] \leq \exp\left( -\frac{P\, t^2}{32\, (c_d + 1)^2} + \frac{1}{4} \right).$$

Thus, we obtain

$$\mathbb{E}[X] = \int_0^\infty \mathbb{P}[X > t] \mathrm{d}t \leq \exp(1/4) \int_0^\infty \exp\left( -\frac{P\, t^2}{32\, (c_d + 1)^2} \right) \mathrm{d}t = \frac{\exp(1/4)\sqrt{32}\pi(c_d + 1)}{2\sqrt{P}},$$

where the last step follows from the identity $\int_0^\infty \exp(-t^2) \mathrm{d}t = \frac{\sqrt{\pi}}{2}$.

Now we proceed to bound the constant $c_d$ in the dimensions. By Theorem 1 we have that $c_d = \frac{\sqrt{\pi}\Gamma(\frac{d+1}{2})}{\Gamma(\frac{d}{2})}$. Now the claim follows from the bound $\frac{\Gamma(\frac{d+1}{2})}{\Gamma(\frac{d}{2})} \leq \sqrt{\frac{d}{2} + \sqrt{\frac{3}{4}} - 1}$ proven by Kershaw (1983). $\square$

## E    COMPARISON OF DIFFERENT KERNELS IN MMD

Next we compare the MMD flows of the Riesz kernels with those of the positive definite kernels

$$K_{\mathrm{G}}(x, y) := \exp\Big( - \frac{\|x - y\|^2}{2\sigma^2} \Big) \quad \text{(Gaussian)},$$

$$K_{\mathrm{IM}}(x, y) := \frac{1}{\sqrt{\|x - y\|^2 + c}} \quad \text{(Inverse Multiquadric)},$$

$$K_{\mathrm{L}}(x, y) := \exp\Big( - \frac{\|x - y\|}{\sigma} \Big) \quad \text{(Laplacian)}.$$

The target distribution is defined as the uniform distribution on three non-overlapping circles and the initial particles are drawn from a Gaussian distribution with standard deviation $0.01$, compare Glaser et al. (2021). We recognize in Figures 4 and 5 that in contrast to the Riesz kernel, the other MMD flows

- heavily depend on the parameters $\sigma$ and $c$ (stability against parameter choice);
- cannot perfectly recover the uniform distribution; zoom into the middle right circles to see that small blue parts are not covered (approximation of target distribution).

Moreover, in contrast to the other kernels, the Riesz kernel with $r = 1$ is positively homogeneous such that the MMD flow is equivariant against scalings of initial and target measure.

Finally, it is interesting that the Riesz kernel is related to the Brownian motion by the following remark, see also Modeste & Dombry (2023) for the more general fractional Brownian motion.

**Remark 13.** *In the one-dimensional case, the extended Riesz kernel with $r = 1$ reads as*

$$K(x, y) = -|x - y| + |x| + |y| = 2\min(x, y),$$

*which is the twice the covariance kernel of the Brownian motion. More precisely, let $(W_t)_{t>0}$ be a Brownian motion and $s, t > 0$. Then, it holds*

$$\mathrm{Cov}(W_s, W_t) = \min(s, t) = \frac{1}{2}K(x, y).$$

## F    TRAINING ALGORITHM OF THE GENERATIVE SLICED MMD FLOW

In Algorithm 3 we state the detailed training algorithm of our proposed method.

## G    ABLATION STUDY

We consider the FID for different number of networks and different number of projections. We run the same experiment as in Section 5 on MNIST. Here we choose a different number of projections $P$ between 10 and 1000. In Figure 6 we illustrate the progress of the FID value for an increasing number of networks and a different number of projections. Obviously, the gradient of the MMD functional is not well-approximated by just using $P = 10$ or $P = 100$ projections and thus the MMD flow does not converges. Once the gradient of the functional is well-approximated, a higher number of projections leads only to a small improvement, see the difference between $P = 500$ and $P = 1000$.

## H    IMPLEMENTATION DETAILS

The code is available online at `https://github.com/johertrich/sliced_MMD_flows`.

We use UNets $(\Phi)_{l=1}^L$ [2] with 3409633 trainable parameters for MNIST and FashionMNIST and 2064035 trainable paramters for CIFAR10. The networks are trained using Adam (Kingma & Ba,

---

[2]modified     from     `https://github.com/hojonathanho/diffusion/blob/master/diffusion_tf/models/unet.py`

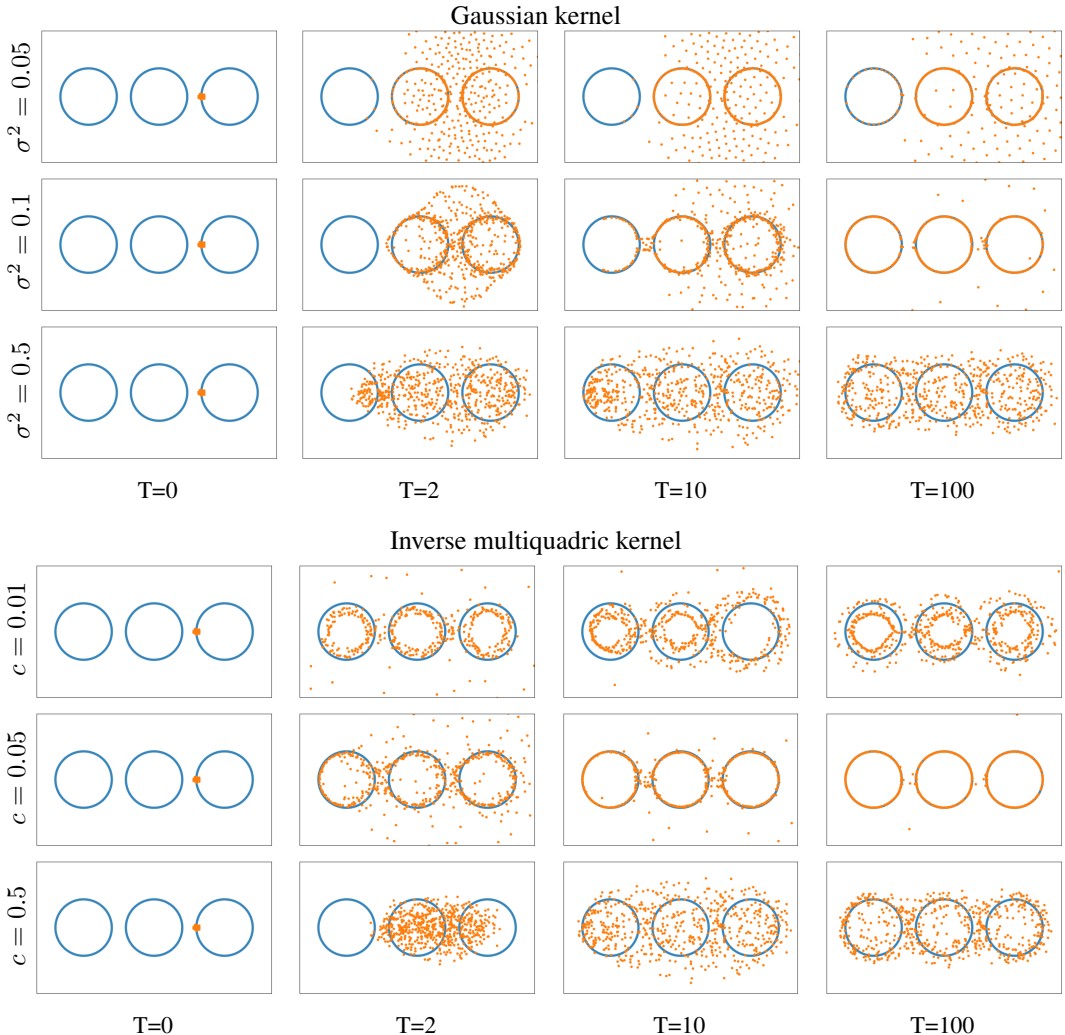

Figure 4: Comparison of the MMD flow with Gaussian kernel (top) and inverse multiquadric kernel (bottom) for different hyperparameters.

2015) with a learning rate of $0.001$. All flows are simulated with a step size $\tau = 1$. We stop the training of our generative sliced MMD flow when the FID between the generated samples and some validation samples does not decrease twice. Then we take the network with the best FID value to the validation set. The validation samples are the last 10000 training samples from the corresponding dataset which were not used for training the generative sliced MMD flow. The training of the generative MMD flow takes between 1.5 and 3 days on a NVIDIA GeForce RTX 2060 Super GPU, depending on the current GPU load by other processes. To avoid overfitting, we choose a relatively small number of optimization steps within the training of the networks $\Phi_l$, which corresponds to an early-stopping technique.

**MNIST.** We draw the first $M = 20000$ target samples from the MNIST training set and $N = 20000$ initial samples uniformly from $[0, 1]^d$. Then we simulate the momentum MMD flow using $P = 1000$ projections for 32 steps and train the network for 2000 optimizer steps with a batch size of 100. After each training of the network, we increase the number of flow steps by $\min(2^{5+l}, 2048)$ up to a maximal number of 30000 steps, where $l$ is the iteration of the training procedure, see Algorithm 3. We choose the momentum parameter $m = 0.7$ and stop the whole training after $L = 55$ networks.

**FashionMNIST.** Here we draw the first $M = 20000$ target samples from the FashionMNIST training set and $N = 20000$ initial samples uniformly from $[0, 1]^d$. Then we simulate the momentum MMD

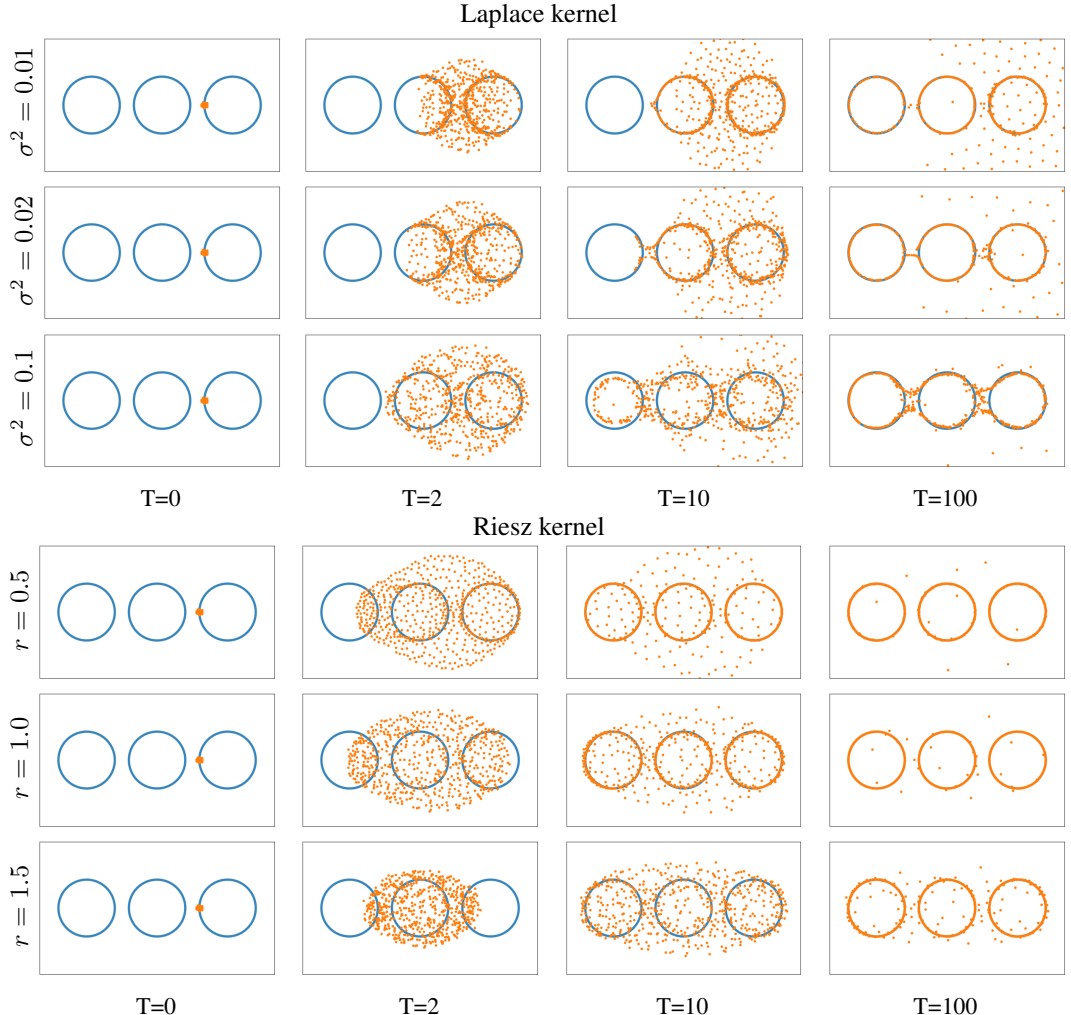

Figure 5: Comparison of the MMD flow with Laplacian kernel (top) and Riesz kernel (bottom) for different hyperparameters.

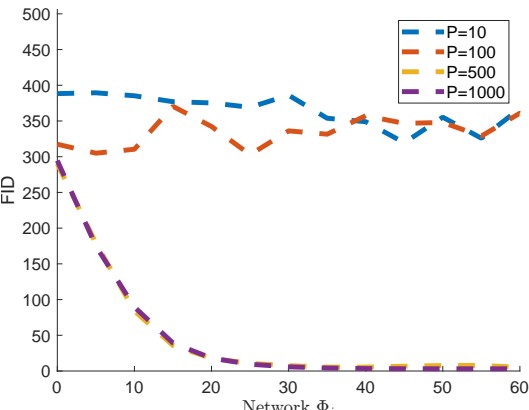

Figure 6: Illustration of the FID value of the Sliced MMD Flow on MNIST for different number of projections.

flow using $P = 1000$ projections for 32 steps and train the network for 2000 optimizer steps with

---

**Algorithm 3** Training of generative MMD flows

---

**Input:** Independent initial samples $x_1^{(0)}, ..., x_N^{(0)}$ from $\mu_0$, momentum parameters $m_l \in [0, 1)$ for $l = 1, ..., L$.
Initialize $(v_1, ..., v_N) = 0$.
**for** $l = 1, ..., L$ **do**
   - Set $(\tilde{x}_1^{(0)}, ..., \tilde{x}_N^{(0)}) = (x_1^{(l-1)}, ..., x_N^{(l-1)})$.
   - Simulate $T_l$ steps of the (momentum) MMD flow:
   **for** $t = 1, ..., T_l$ **do**
     - Update $v$ by

$$(v_1, ..., v_N) \leftarrow \nabla F_d(\tilde{x}_1^{(t-1)}, ..., \tilde{x}_N^{(t-1)} | y_1, ..., y_M) + m_l(v_1, ..., v_N)$$

     - Update the flow samples:

$$(\tilde{x}_1^{(t)}, ..., \tilde{x}_N^{(t)}) = (\tilde{x}_1^{(t-1)}, ..., \tilde{x}_N^{(t-1)}) - \tau N (v_1, ..., v_N)$$

   **end for**
   - Train $\Phi_l$ such that $\tilde{x}^{(T_l)} \approx \tilde{x}_i^{(0)} - \Phi_l(\tilde{x}_i^{(0)})$ by minimizing the loss

$$\mathcal{L}(\theta_l) = \frac{1}{N} \sum_{i=1}^{N} \|\Phi_l(\tilde{x}_i^{(0)}) - (\tilde{x}_i^{(0)} - \tilde{x}_i^{(T_l)})\|^2.$$

   - Set $(x_1^{(l)}, ..., x_N^{(l)}) = (x_1^{(l-1)}, ..., x_N^{(l-1)}) - (\Phi_l(x_1^{(l-1)}), ..., \Phi_l(x_N^{(l-1)}))$.
**end for**

---

a batch size of 100. After each training of the network, we increase the number of flow steps by $\min(2^{5+l}, 2048)$ up to a maximal number of 50000 steps, where $l$ is the iteration of the training procedure. The momentum parameter $m$ is set to 0.8. We stop the whole training after $L = 67$ networks.

**CIFAR.** We draw the first $M = 30000$ target samples from the CIFAR10 training set. Here we consider a *pyramidal schedule*, where the key idea is to run the particle flow on different resolutions, from low to high sequentially. First, we downsample the target samples by a factor 8 and draw $N = 30000$ initial samples uniformly from $[0, 1]^{\frac{d}{64}}$. Then we simulate the momentum MMD flow in dimension $d = 48$ using $P = 500$ projections for 32 steps and train the network for 5000 optimizer steps with a batch size of 100. After each training of the network, we increase the number of flow steps by $\min(2^{5+l}, 1024)$ up to a maximal number of 30000 steps, where $l$ is the iteration of the training procedure. The momentum parameter $m$ is increased after each network training by 0.01 up to 0.8, beginning with $m = 0$ in the first flow step. We increase the resolution of the flow after 600000 flow steps by a factor 2 and add Gaussian noise on the particles in order to increase the intrinsic dimension of the images, such that the second resolution is of dimension $d = 192$. Following here the same procedure as before for the second resolution and the third resolution of $d = 768$, we change the projections in the final resolution of $d = 3072$. Instead of using projections uniformly sampled from $\mathbb{S}^{d-1}$, we consider *locally-connected* projections as in (Du et al., 2023; Nguyen & Ho, 2022). The idea is to extract patches of the images $\boldsymbol{x}^k$ at a random location in each step $k$ and instead consider the particle flow in the patch dimension. In order to apply these locally-connected projections at different resolutions, we also upsample the projections to different scales. Here we choose a patch size of $7 \times 7$ and consider the resolutions $7, 14, 21, 28$. We stop the whole training after $L = 86$ networks.

Note that herewith we introduced an inductive bias, since we do not uniformly sample from $[0, 1]^d$, but empiricially this leads to a significant acceleration of the flow. A more comprehensive discussion can be found in (Du et al., 2023; Nguyen & Ho, 2022)

**CelebA.** We draw the first $M = 20000$ target samples from the CelebA training set. Again, we consider a pyramidal schedule as for CIFAR10, but we increased the number of flow steps by $\min(2^{5+l}, 8192)$ up to a maximal number of 100000 steps. We increase the resolutions of the flow after 700000, 900000 and 700000 flow steps by a factor 2 and add Gaussian noise on the particles in order to increase the intrinsic dimension of the images. We also use locally-connected projections

Table 2: FID scores of generated samples for training set and test set

|  | MNIST | FashionMNIST | CIFAR10 | CelebA |
|---|---|---|---|---|
| training set | 2.7 | 10.6 | 53.0 | 32.7 |
| test set | 3.1 | 11.3 | 54.8 | 32.1 |

with a patch size of $7 \times 7$ and consider the resolutions $7, 14, 21, 28$ for resolution $32 \times 32$ and $7, 14, 21, 28, 35, 42, 49$ for resolution $64 \times 64$. We stop the whole training after $L = 71$ networks.

## I ADDITIONAL EXAMPLES

In Figure 7 we show more generated samples of our method for MNIST, FashionMNIST and CIFAR10.

In Figure 8, we compare generated MNIST samples with the closest samples from the training set. We observe that they are significantly different. Hence, our method generates really new samples and is not just reproducing the samples from the training set. In contrast, in Figure 9 we compare the particle flow samples with the closest samples from the training set. Obviously, the samples of the particle flow approximate exactly the training samples. This highlights the important role of the networks: We can interpolate between the training points in order to generalize the dataset.

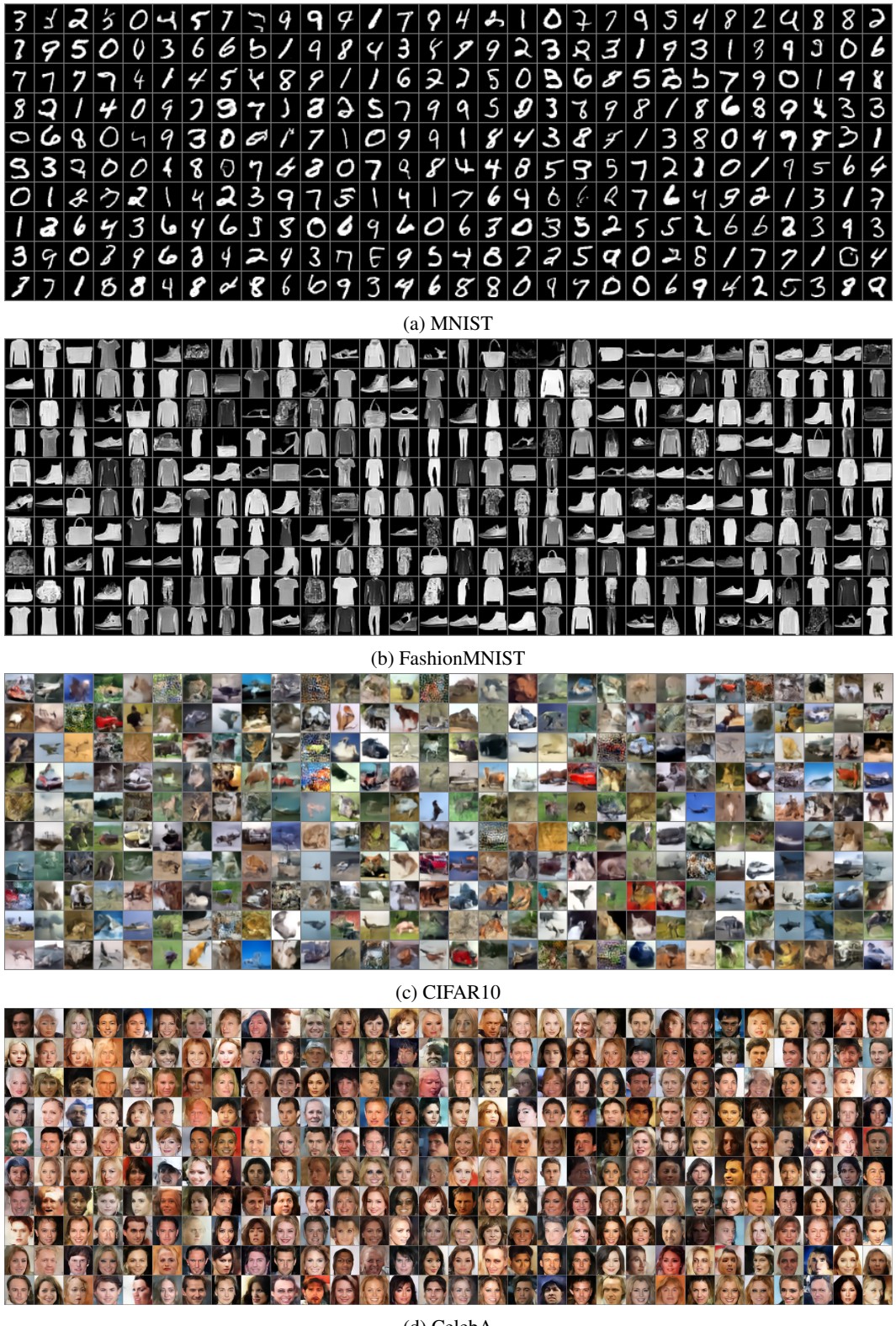

(a) MNIST

(b) FashionMNIST

(c) CIFAR10

(d) CelebA

Figure 7: Additional generated samples (from top to bottom) of MNIST, FashionMNIST, CIFAR10 and CelebA.

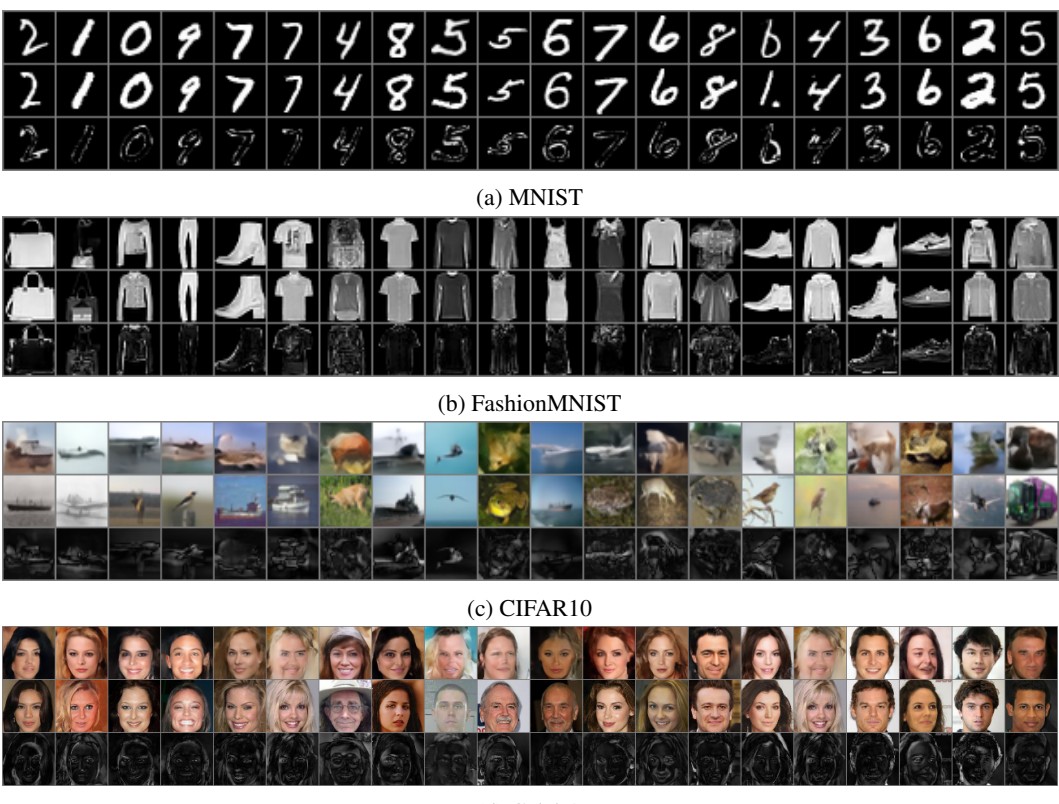

Figure 8: Generated samples (top), $L_2$-closest samples from the training set (middle) and the pixelwise distance between them (bottom).

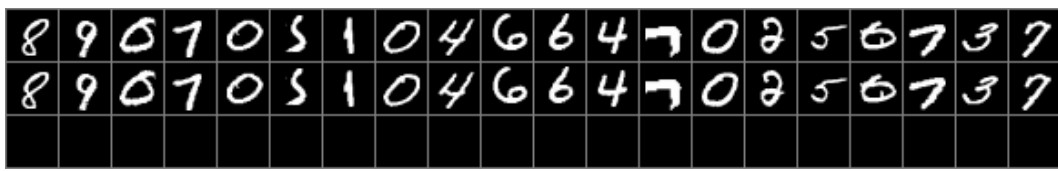

Figure 9: Samples of the particle flow (top), closest samples from the training set (middle) and the pixelwise distance between them (bottom). The mean PSNR between these flow samples and the corresponding closest training images is 82.29.

