# OpenReview forum: "Generative Sliced MMD Flows with Riesz Kernels"
_ICLR.cc/2024/Conference — ICLR 2024 poster_

### Official Review · Reviewer_HuoH · 2023-10-18

**Soundness:** 3 good
**Presentation:** 3 good
**Contribution:** 3 good
**Rating:** 6
**Confidence:** 3

**Summary:**

The paper proposes a non-parametric generative model, sliced MMD Flows with Riesz Kernels. The paper initially presents the concept of "Sliced MMD with Riesz Kernels," which is essentially a sliced variation of MMD with Riesz Kernels. The paper demonstrates that Sliced MMD with Riesz Kernels qualifies as a metric in the space of probability distributions, having non-negativity, symmetry, triangle inequality, and identity. Additionally, it establishes an equivalence between Sliced MMD with Riesz Kernels and the MMD with sliced Riesz Kernels. Furthermore, the paper elaborates on the methodology for calculating Sliced MMD with Riesz Kernels and its gradient, showcasing that this approach achieves nearly linear complexity in relation to the support count of two discrete distributions. Lastly, the paper compares the proposed frameworks with other generative modeling techniques, such as NCSN, WGAN, MMD GAN, SIG, SWF, and more.

**Strengths:**

* The paper represents a progression in utilizing MMD flows for generative modeling. Through the introduction of an innovative and clever method for calculating gradients of MMDs using Riesz kernels, the suggested approach opens up the possibility of employing MMD flows in generative modeling tasks that were previously considered impractical for these types of functions.
* The demonstration of the metric property and calculating MMD gradients linked to Riesz kernels is a great contribution. it brings attention to Riesz kernels, which might have been somewhat neglected within the broader landscape of kernel methods.
* The connection with Wasserstein distance is interesting.
* The paper archives the best FID score on MNIST and Fashion MNIST.

**Weaknesses:**

* The FID score on Cifar10 is relatively high to other generative models.
* The computation of the FID score is only from 1 run without any standard deviation.
* It seems that the proposed framework is not scalable in terms of dimension since the result from CIFAR10 is quite blurry and noisy.

**Questions:**

* Could any methods be used to improve the experimental results on high-dimensional datasets?

---

> ### Author Response · Authors · 2023-11-16
> **Rebuttal**
>
> Thank you very much for your review.
> We extended the experimental part by including the CelebA dataset which is of higher dimension than the other datasets.
>
> In order to circumvent the dimensionality effect, we use on CelebA two further techniques.
>
> - We use the multiscale approach to generate the images. More precisely, we first generate a downsampled version of the CelebA dataset, use an upsampling step and approximate the higher-dimensional version.
> From a theoretical side this correspond to change the latent distribution.
>
> - Following the references (Du et al. 2023) and (Nguyen and Ho, 2022), we included local projections.
>
> We describe both techniques in more detail within the implementation details setting in Appendix H.
>
> To explain the dependence on the dimensions theoretically, we have already included Remark 5 in the main paper.
>
> References:
>
> (Du et al, 2023) Nonparametric Generative Modeling with Conditional Sliced-Wasserstein Flows
>
> (Nguyen and Ho, 2022) Revisiting sliced Wasserstein on images: From vectorization to convolution

---

> > ### Comment · Reviewer_HuoH · 2023-11-20
> > **Response to the authors**
> >
> > Thank you for adding new results to the CelebA dataset. Nevertheless, the generative quality of the proposed flow is not as good as current generative modeling literature e.g., Table 1 and 2 [1]. Therefore, I will keep my initial score.
> >
> > [1] GENERATIVE MODELING WITH OPTIMAL TRANSPORT MAPS, Litu et al

---

### Official Review · Reviewer_hi5p · 2023-10-31

**Soundness:** 3 good
**Presentation:** 2 fair
**Contribution:** 3 good
**Rating:** 8
**Confidence:** 4

**Summary:**

This article focuses on MMD flows with a Riesz kernel that is the distance between points raised to a power $r \in (0,2)$. The main contributions are, firstly, to demonstrate that this kernel is identical (up to a constant) to its 'sliced' version and, secondly, to use this characterization for the efficient computation of gradients in MMD flows.

**Strengths:**

I find this article to be well-written and its contributions to be interesting. Efficient MMD computation is indeed an important point, not only for MMD flows. The article addresses an important problem and offers an elegant solution for Riesz kernels. However, there are several points that appear to need correction or, at the very least, further elaboration.

**Weaknesses:**

- Concerning Theorem 2:

Theorem 2 establishes bounds between MMD and Wasserstein distance of order 1. In my opinion, these results are not very sharp, and there appears to be an important missing reference here. Under the same assumptions of compact support, the article [1, Theorem 1] demonstrates that the Wasserstein distance $W_1$ is bounded by an MMD with the Coulomb kernel $k(x, y) = -|x - y|^{2-d}$ but without the power dependency of $1/(d+1)$. Since the measures are bounded, MMD with the Riesz kernel and the Coulomb kernel are related by a constant (dependent on the dimension) on the support of the measures. Therefore, it seems important to mention this result and discuss its relevance.

In a more general sense, I'm having trouble grasping the significance of these bounds for the current article. They don't appear to be entirely novel, and more importantly, they are not utilized subsequently, neither theoretically nor practically, nor in the discussion.

- Regarding Theorem 3:

Theorem 3 is the main contribution of this article, and I agree that it is interesting and significant. However, there is a minor point that needs clarification: while the function $E$ is shown to be differentiable everywhere in the proof, for the differentiability of $V$, the authors use an argument that doesn't seem rigorous. Indeed, the function $x \to |x-y|$ is not differentiable at $x=y$. In practice, this may not be very important because this event is almost surely zero if the samples come from measures with densities. Still, it's important to note this limitation.

- Regarding the dimension dependency:

An important aspect that is not detailed in the article is the dependence on dimension. Indeed, the sliced Riesz Kernel is not exactly equal to the Riesz kernel; it differs by a constant that depends on the dimension. As the dimension approaches infinity, this constant $c_{d,r}$ behaves as $O(d^{r/2})$, and thus the rescaled kernel tends to zero as $O(d^{-r/2})$. This suggests that estimating the gradient with a finite number of projections becomes increasingly challenging in high dimensions, as also observed in Figure 2 and the bound in $O(\sqrt{d/P})$.

This point is not thoroughly discussed in the article. It would be interesting to visualize the relative error in the gradient not only as a function of the number of projections but also with respect to the dimension.


- About training a sequence of neural networks (Section 4.2):

I am having difficulty grasping the intuition behind the iterative training of neural networks $\Phi_1, \cdots, \Phi_L$ to approximate the generation scheme. Why are these networks needed? Does the flow not work without them? What are the results without these networks on CIFAR10 or MNIST, for example?

Furthermore, it seems to me that training a network per step is very costly; I doubt the feasibility of this method. Is this standard practice?

- Regarding the "related work" section in the introduction:

I believe that this section could be improved. The paragraph is somewhat confusing as it introduces a set of articles without providing clear context for the current work or establishing connections between the cited articles. As a result, I find this part to be not very illuminating for understanding the related work.


- Additional remarks:

  - Figures 1 and 2: The tables are not informative and hard to read. Since the information is already contained in the plots, I don't see the purpose of these tables. I suggest removing them and using the available space for more details on the points described earlier. Additionally, Figure 1 lacks a legend for the runtime (in ms?).

  - The Riesz kernel also defines a valid MMD for $r=2$. So why restrict it to $r \in (0,2)$ with 2 excluded?


  - Unless I am mistaken, the result about $D_{K}$ as a metric on $P_{r/2}$ in (Modeste & Dombry, 2023) is not mentioned anywhere in the cited article. Can you provide more details on this fact or provide the correct reference ?

  - I'm quite curious to know if the approach presented in the article can be generalized to the case of $r \in (0,2]$? Particularly, the fast gradient computation.

  - It seems that the reference (Numayer & Steidl, 20201, Lemma 3.3) is incorrect, or at least Lemma 3.3 doesn't state that the two MMDs coincide.

- Refs:

[1] Djalil Chafaï, A. Hardy, and Mylène Maïda. Concentration for Coulomb gases and Coulomb transport inequalities. Journal of Functional Analysis, 275, 2016.

**Questions:**

see above

---- AFTER REBUTTAL ----

The authors have addressed all my concerns.

---

> ### Author Response · Authors · 2023-11-16
> **Rebuttal part 1 of 2**
>
> Many thanks for your thorough and detailed review. We answer to all of your comments separately.
>
> - Concerning Theorem 2:
> the statement of Theorem 2 does not follow from a corresponding statement about the Coulomb kernel as we explain below.
> The Coulomb kernel in Chafaï et al. is defined by $K_C(x,y)=\\|x-y\\|^{2-d}$ for $d\ge3$ **without the minus**.
> Therefore, we cannot bound the Riesz kernel from below by the Coulomb kernel.
> Moreover, Thm 1.1 from Chafaï et al. assumes that $\int\int K_C(x,y)d \mu(x)d \mu(y)<\infty$ and $\int\int K_C(x,y)d\nu(x)d \nu(y)<\infty$.
> This assumption is violated whenever $\mu(\\{x\\})>0$ or $\nu(\\{x\\})>0$ for any $x$.
> In particular, this assumption excludes the case of empirical measures which is the most important one for our applications.
> Nevertheless, since this is an interesting idea and since the bound looks similar at the first glance, we added a comment about that at the end of the proof of Theorem 2 in Appendix B. Even though we do not utilize this bound, we found the relations to the Wasserstein-1 distance quite interesting and would like to keep it in the paper.
>
> - Concerning the sharpness of the bounds in Theorem 2:
> We do not think that a sharper version of Theorem 2 without the dimension scaling holds true for the following reason:
> It is well-known that MMD has sample complexity $O(n^{-1/2})$ (Gretton et al. 2012) and Wasserstein-1 has a sample complexity of $O(n^{-1/d})$ (see e.g., Peyré and Cuturi, 2020). If it would hold true that $W_1(\mu,\nu)\leq C\mathcal D_K(\mu,\nu)$ for all $\mu,\nu$ this would imply that also Wasserstein-1 would have a sample complexity of $O(n^{-1/2})$ which is a contradiction.
> We included a comment about that in the paper.
> For the Coulomb kernel this argumentation does not apply, since it is not bounded on compact sets (which is an assumption for the sample complexity $O(n^{-1/2})$ for MMD) and since the result of Chafai et al. explicitly excludes empirical measures.
>
> - Regarding Theorem 3: Note that the potential energy $V$ is convex.
> Consequently, we can use subgradients instead of gradients. We added a comment on this in the paper after Theorem 3.
>
> - Regarding the dimension dependency:
> We thank the reviewer for raising this point. The fact that $c_{d,r}\in O(d^{r/2})$ is exactly the reason why the $\sqrt{d}=d^{1/2}=d^{r/2}$ appears within the error bound for the gradients in Theorem 4, where $r=1$.
> We already discussed the consequences in Remark 5.
> In particular, this causes that the computational complexity for the estimation of gradients of the MMD up to a fixed error $\epsilon$ via slicing depends quadratically on $d$ and not linearly.
> In the updated version, we inserted comments after Theorem 1 and in Remark 5 in order to make this point more clear.
> Moreover, we inserted a plot with the approximation error on the right of Figure 1.
>
> - About training a sequence of neural networks (Section 4.2):
> We train the networks for two reasons.
> First, the neural networks speed up the generation process and allow us to draw a single sample (for the particle flow, we always have to compute a batch of samples).
> The second and more important reason is that the particle flow itself converges **exactly** to the training images, see Figure 9 and the corresponding explanation in the Appendix I of the paper for an illustration.
> In some sense, we could say that the particle flow works "too well" since it approximates the discrete target measure exactly. The resulting FID of the flow particles would be 1.37 for MNIST, 1.90 for FashionMNIST and 3.90 for CIFAR10.
> If we use the sequence of neural networks, we can interpolate between the training points in order to generalize the dataset. Finally, note that we do not train a network per step of the particle flow, but one network for 30000 steps of the flow after a warm start procedure. A detailed explanation towards this is given in Appendix H. Thus the training of the networks is not very costly compared to the simulation of the particle flow.
>
> - Regarding related work:
> We agree that several references from the related work section were relevant for providing a broader context of the topic, but were not directly related to our paper. We moved these references to the general introduction and reorganized the related work section.
> We hope that this makes it more clear now.

---

> > ### Author Response · Authors · 2023-11-16
> > **Rebuttal part 2 of 2**
> >
> > ## Other comments
> >
> > - We followed your suggestion to fuse the graphs from Figure 1 and 2 and omit the tables. Moreover, we included the legend for the runtime, it is in fact in ms.
> >
> > - If we use $r=2$ in the Riesz kernel, the corresponding MMD is no longer a metric. Indeed it holds that
> > $$
> > \mathcal D_K^2(\mu,\nu)=\frac12\int\int-\\|x-y\\|^2 \mathrm{d} (\mu-\nu)(x) \mathrm{d} (\mu-\nu)(y)
> > $$
> > $$
> > =\sum_{i=1}^d \frac12\int\int-(x_i-y_i)^2 \mathrm{d} (\mu-\nu)(x) \mathrm{d} (\mu-\nu)(y) =\sum_{i=1}^d\mathcal D_K^2((\pi_i)\\#\mu,(\pi_i)\\#\nu),
> > $$
> > where $\pi_i(x)=x_i$. In particular, we have for $K(x,y)=-\\|x-y\\|^2$ that $\mathcal D_K^2(\mu,\nu)$ is zero whenever the marginals $(\pi_i)\\#\mu$ and $(\pi_i)\\#\nu$ coincide for all $i$. Consequently, $\mathcal D_K$ cannot be a metric in this case.
> > We added a comment about that in the paper.
> >
> > - Since the statement about the metric property of $\mathcal D_{\tilde{K}}$ on $\mathcal P_{\frac r 2}(\mathbb R^d)$ is not directly relevant in the main paper, we moved it to Remark 7 in Appendix A. Moreover, we added the following explanations, how it can be deduced from Proposition 2.14 in (Modeste and Dombry 2023):
> > By Theorem 4.26 in (Steinwart and Christmann, 2008), the MMD
> > $$
> > \mathcal D_{\tilde K}^2(\mu,\nu)=\int\int \tilde K(x,y) d(\mu-\nu)(x)d(\mu-\nu)(y)
> > $$
> > is finite on $\mathcal M_{\tilde K}=\\{\mu \in \mathcal M(\mathbb R^d):\int \sqrt{\tilde K(x,x)}\mathrm d |\mu|<\infty\\}$, where $\mathcal M(\mathbb R^d)$ is the space of all signed measures on $\mathbb R^d$.
> > Since $\tilde K(x,x)=2\\|x\\|^r$, we have that $\mathcal M_{\tilde K}=\\{\mu \in \mathcal M(\mathbb R^d):\int \\|x\\|^{\frac r 2}\mathrm d |\mu|<\infty\\}$ such that $\mathcal M_{\tilde K}\cap\mathcal P(\mathbb R^d)=\mathcal P_{\frac r 2}(\mathbb R^d)$.
> > Now, inserting $\tilde K$ Proposition 2.14 in (Modeste and Dombry 2023) states that $\mathcal D_{\tilde{K}}$ is a metric on $\mathcal M_{\tilde K}\cap \mathcal P(\mathbb R^d)=\mathcal P_{\frac r 2}(\mathbb R^d)$, where the assumptions of the proposition are checked in Example 2 and 3 of (Modeste and Dombry 2023) and $\tilde K$ is named $k_H$ with $H=\frac r 2$.
> >
> > - While Theorem 2 works with any $r\in(0,\infty)$ (even though we loose properties of the MMD for $r\geq 2$), we are quite sure that the one-dimensional sorting algorithm cannot be generalized to other cases. However, we are currently working on a generalization of this approach via Fourier transforms and hope to finish a paper on it until the end of the year.
> >
> > - We are sorry about the confusion regarding the reference (Neumayer and Steidl, 2021). We just recognized that the numbering of the lemmas differs between the arxiv and journal version.
> > The statement which we refer to is part iii) of Lemma 3.3 in the arxiv version and part iii) of Lemma 1 in the journal version.
> > We corrected the reference in the paper.
> >
> > References:
> >
> > (Gretton et al., 2012) A kernel two-sample test
> >
> > (Peyré and Cuturi, 2020) Computational optimal transport
> >
> > (Neumayer and Steidl, 2021) From Optimal Transport to Discrepancy
> >
> > (Steinwart and Christmann, 2008) Support vector machines
> >
> > (Modeste and Dombry, 2023)  Characterization of translation invariant MMD on $\mathbb R^d$ and connections with Wasserstein distances
> >
> > (Chafaï et al., 2016) Concentration for Coulomb gases and Coulomb transport inequalities.

---

> > ### Comment · Reviewer_hi5p · 2023-11-16
> > **Reponse to comment 1**
> >
> > Thank you very much for your reponse.
> >
> > For the first point indeed I was wrong, I missed the minus, thank you for pointing that. However it is not clear (and I am not sure I agree) why the assumptions of Chafaï "exclude the case of empirical measures". Unless I am missing something for any empirical measure $\mu= \frac{1}{n} \sum_{i} \delta_{x_i}$ where $x_i \in \Omega$ for compact $\Omega$ then we have $\int \int K_C(x,y) d\mu(x) d\mu(y) < +\infty$ and result from Chafaï applies. Also if all the distributions belong to some low-dim space we could have better sample complexity and the bound $W \leq MMD$ without the dependence in $d$ (or with a better one) in the exponent (see for example Section 2.4 in [1])
> >
> > My remaining point is the neural network strategy for learning the ‘‘generative flow". I am still doubting about this strategy, as it seems not standard. From my understanding of generative particle flows; when we have a discrepancy $F$ between probability distributions, to make a generative model what is usually done is something as computing a particle flow on $F(NN \circ \mu_Z, \nu)$ where $\mu_Z$ is a simple latent distribution and $NN$ a neural network. Then to sample you only have to generate from $\mu_Z$.
> >
> > For what I understand your strategy is to train NNs to approximate $T_l$ steps of the flow, so that the flow with these NN is not exact + it allows to generate for unseen point. Do you have any references that also use this strategy ?
> >
> > The drawback I see is that the neural network could easily overfit the data so that the generated data are actually very similar to training data, thus causing a severe a memorization issue. It seems actually the case in Figure 8 where the generated samples are very similar to training points and not very diverse. Also as described in [2] just looking at nearest neighbors for assessing the diversity of the generated samples is not a very good option (the of only few different pixels can easily ‘‘fool'' the Euclidean metric) so that I'm not really convinced by this Figure 8.
> >
> > As you commented the FID scores are much smaller for the particle flows (so ‘‘better scores‘‘ according to the FID), suggesting this memorization issue occurs for the particle flows. It is indeed known that FID score assign a very good score to a model which simply memorizes small samples from the true distribution [3]. But I really doubt that your approach with NN actually solves this issue, as there is no ‘‘randomness'' involved in the model.
> >
> > [1] Controlling Wasserstein Distances by Kernel Norms with Application to Compressive Statistical Learning, Titouan Vayer and Rémi Gribonval.
> > [2] On Memorization in Probabilistic Deep Generative Models, Gerrit J.J. van den Burg, Christopher K.I. Williams.
> > [3] Towards gan benchmarks which require generalization, Ishaan Gulrajani, Colin Raffel, and Luke Metz.

---

> > > ### Author Response · Authors · 2023-11-17
> > > **Response**
> > >
> > > Thanks for the quick reply! Please find below some additional explanations on your questions. Additionally, we added more experiments regarding the generalization/memorization in Figure 8 in the appendix to underline our findings numerically.
> > >
> > > ## Regarding the Paper (Chafaï et al., 2016)
> > >
> > > - In the definition of the Coulomb kernel, (Chafaï et al., 2016) consider only the case $d\ge 2$. For $d=2$, the Coulomb kernel is defined by $K_C(x,y)=\log \frac{1}{\\|x-y\\|} $ and for $d\ge 3$ by $K_C(x,y)= \frac{1}{\\|x-y\\|^{d-2}}$. Note that in both cases, $K_C$
> > > is only well-defined for $x\neq y$ and if we deal with an empirical measure (take $\mu=\delta_0$ for simplicity),
> > > we have to care how we extend the kernel for $x=y$. If we define $K_C(x,x)=+\infty$, then we have that
> > > $\int\int K_C(x,y)\mathrm{d}\delta_0(x)\mathrm{d}\delta_0(y)=K_C(0,0)=+\infty$ and the same happens for any other empirical measure.
> > >
> > > - In fact, (Chafaï et al., 2016) explicitly use that $\int\int K_C(x,y)d\mu(x)d\mu(y)=\infty$ for empirical measures:
> > > In the proof of Theorem 1.1 on page 16 (arxiv version) (Chafaï et al., 2016) state that $\mathcal E(\mu,\nu)<\infty$ implies $\mu\otimes\nu(\\{x=y\\})=0$, where $\mathcal E(\mu,\nu)=\int\int K_C(x,y)d\mu(x)d\nu(y)$.
> > > Inserting $\mu=\nu$ and noting that $\mu\otimes\mu(\\{x=y\\})=0$ implies that $\mu(\\{x\\})=0$ for all $x\in\mathbb R^d$, we obtain that
> > > $\int\int K_C(x,y)d\mu(x)d\mu(y)<\infty$ implies that $\mu(\\{x\\})=0$ such that $\mu$ cannot be an empirical measure.
> > >
> > > - We agree that you might get sharper bounds in less general settings, e.g., if you restrict the set of the considered probability measures to be supported on low-dimensional subspaces (which seems to be quite restrictive).
> > > However, the statement of the theorem is about **all** probability measures supported on $B_R(0)$.
> > > We still do not think that there is a **general** dimension-independent bound.
> > >
> > > ## Generalization
> > >
> > > - We have randomness both in the training as well as in the sampling process. In the training process, the initial samples are drawn randomly from the latent distribution $P_Z$. Afterwards, we train the $\Phi_l$ based on the corresponding particle flow.
> > > Here, the main purpose of the networks is to learn a meaningful interpolation between the sample paths.
> > > Once the $\Phi_l$ are trained, we generate samples exactly in the way which you suggested: We sample $z$ from $P_Z$ and compute then $NN(z)$, where $NN=\Phi_L\circ\cdots\circ\Phi_1$.
> > > Since $z$ was none of the initial samples from the training process (with probability 1), the model will generate a sample which was not contained in the training dataset.
> > >
> > > - We approximate the Wasserstein gradient flow $\gamma\colon[0,\infty)\to\mathcal P_2(\mathbb R^d)$ with $\gamma(0)=P_Z$ wrt to MMD.
> > > Now, $\Phi_l$ is trained such that $\gamma(t_l)\approx \Phi_l\\#\gamma(t_{l-1})$ for some $0=t_0<t_1<\cdots<t_L$.
> > > Then it holds $\gamma(t_L)\approx(\Phi_L\circ\cdots\circ\Phi_1)\\#\gamma(0)$ with $\gamma(0)=P_Z$.
> > > Such methods learning iteratively an "interpolation path" are exploited several times in literature, e.g.,
> > >
> > > Arbel et al. 2021, Annealed flow transport Monte Carlo
> > >
> > > Fan et al. 2022, Variational Wasserstein Gradient Flow
> > >
> > > Ho et al. 2020, Denoising diffusion probabilisitc models
> > >
> > > - Of course, we have to avoid overfitting when learning the networks $\Phi_l$. To this end, one can apply standard techniques.
> > > In our case, we implement early-stopping by using a relatively short training time of the networks (e.g. 2000 optimizer steps with batch size 100 for (Fashion)MNIST).
> > > We want to emphasize that for most generative models (including GANs and normalizing flows) such an implicit regularization by the architecture or training procedure is necessary.
> > >
> > > - The MNIST dataset is very dense. If you draw a MNIST-like handwritten digit it will be very close to one of the training samples with high probability.
> > > In fact, the diversity in Fig 8 is quite similar as for standard generative models like GANs (see e.g. Fig 2b in Goodfellow et al., 2014, Generative adversarial networks).
> > > We have now extended Fig 8 for the other datasets. Already for FashionMNIST the differences are larger and for CIFAR10 and CelebA there is only a loose similarity between generated images and the $L^2$-nearest neighbor.
> > >
> > > - Additionally we added an experiment, which computes the FID between the generated samples and the training dataset as well as between the generated samples and the test dataset.
> > > As we can see, there is a very small difference, meaning that we do not have a significant overfitting effect.
> > >
> > > - We agree that the FID as well as $L^2$-nearest neighbors have significant weaknesses as evaluation metrics. However, these methods are common-practice in generative modelling and we are not aware of evaluation metrics without drawbacks.

---

> ### Comment · Reviewer_hi5p · 2023-11-17
> **Response to all the comments**
>
> About (Chafaï et al., 2016): you are absolutely correct and I missed that point, this is a good remark. For the generalization/memorization discussion I thank you for the answers. I think it would be good to describe this in the paper: especially the formalism with the push-forward operator and the interpolation path (which were enlightening; at least for me).
>
> All my concerns are addressed and I change my score accordingly.

---

> > ### Author Response · Authors · 2023-11-20
> > **Response**
> >
> > We would like to thank the reviewer once again for his useful comments. We included the description of the pushforwards and the "interpolation path" in Section 4.2 and the description of the early-stopping in Appendix H.

---

### Official Review · Reviewer_pBbs · 2023-10-31

**Soundness:** 2 fair
**Presentation:** 3 good
**Contribution:** 3 good
**Rating:** 5
**Confidence:** 2

**Summary:**

The paper is dedicated to the question: how to estimate a gradient of the MMD distance between two empirical distributions w.r.t. points of the first distribution. The MMD distance is a natural distance between distributions, using which one can solve generative modeling problems. So the latter computational problem is quite important for generative modeling based on kernels. It is shown that the MMD distance defined by the Riesz kernel has a very special structure, namely that the so-called sliced Riesz distance coincides with the MMD distance.
This allows one to estimate the gradient very precisely because one can take few 1-dimensional projections of empirical points and calculate an averaged sliced MMD. Based on that, authors design a generative modeling algorithm (described in the Appendix as Algorithm 3). Their algorithm performs well, taking into account Table 1, though which part of their algorithm is mainly responsible for such a promising outcome is a non-trivial issue.

**Strengths:**

Major theoretical claims are correct, and proofs seem convincing, though I have not checked all of them.

**Weaknesses:**

The paper is dedicated to accelerating the computation of the gradient of the sliced MMD with the Riesz kernel. Experiments are dedicated to a new algorithm for generative modeling (Algorithm 3 described in Appendix). A natural question appears: what is responsible for good results on MNIST/FashionMNIST/CIFAR10? Is it the sequential approach to train MMD flows, or the fact that gradients are estimated better, or the fact that Riesz kernel defines such a special MMD, or maybe specifics of architecture of neural networks Ф_1, ..., Ф_L (modified from some previous work)?

For me, it is hard to make a judgment of what these experimental results really mean. There are too many ingredients there.

**Questions:**

A natural question: is the Riesz kernel so special, that the MMD distance induced by it leads to successful generative modeling? Or your algorithm for an accurate approximation of gradient is responsible for success?

---

> ### Author Response · Authors · 2023-11-16
> **Rebuttal**
>
> Thank you very much for your comments. We want to emphasize some advantages of our method which explain why it works that well.
>
> - The MMD and the energy distance (= MMD with Riesz kernel and $r=1$) have shown to be very effective two-sample tests to compare high-dimensional probability measures, see (Szekely, 2004). Therefore, minimizing these metrics via gradient flows is a natural approach. However, previous works on MMD flows (e.g. Arbel et al, 2019) were only applicable for small problems due to the quadratic computational cost in the number of samples. Only with our reduction of the computational cost it is possible to compute such MMD flows on datasets with 50000 images or more in a reasonable time.
>
> - Accurate gradient evaluations: We have proven convergence guarantees stating that the sliced MMD gradients converge to the original MMD gradients.
> Moreover, we do not require to solve an optimization problem within the steps which is necessary for other gradient flow methods which are often based on implicit schemes (like JKO).
>
> - The sequential learning approach is computationally efficient.
> The whole generative model $\Phi_L\circ\cdots\circ\Phi_1$ has a large number of parameters and is (as usual for generative models) a highly irregular mapping. However, during the training we only have to consider one of the networks $\Phi_l$ at a time. This leads to a sequence of easier learnable problems instead of one hard one.
>
> We added a paragraph "Discussion" in the conclusions section which summarizes these points.
>
> For the architecture of the $\Phi_l$ we used a standard UNet with the implementation of (Huang et al. 2021). Moreover, in our experiments changes in the architecture did not have a large effect on the results as long as it is expressive enough.
>
> References:
>
> (Szekely et al, 2004) Testing for equal distributions in high dimension
>
> (Arbel et al, 2019) Maximum Mean Discrepancy Gradient Flow
>
> (Huang et al, 2021) A variational perspective on diffusion-based generative models and score matching

---

### Official Review · Reviewer_e9Tn · 2023-11-01

**Soundness:** 2 fair
**Presentation:** 3 good
**Contribution:** 2 fair
**Rating:** 5
**Confidence:** 3

**Summary:**

The work proposed to use sliced MMD with Riesz kernel  to compute MMD gradients for generative model training. The authors introduced the Riesz kernel with its sliced version in section 2 and show that  sliced version is actually the Riesz kernel. Section 3 showed how to compute gradients of sliced MMD in one-dimensional space by its special property of ordering projected data. Section 4 presented MMD flows. The author demonstrated their methods in section 5 with MNIST, FashionMNIST and CIFAR10 datasets.

**Strengths:**

The paper is easy to follow and read.
The proposed method is simple and computational efficient.
The experiment results showed an improvement of FID in MNIST and FashionMNIST data sets.

**Weaknesses:**

All the theory part is quite simple, specially the important theorem 1, which proved that the Sliced Riesz kernel is an equivalent form of Riesz kernel. I have the same impression for the sorting algorithm in 1-D case and results of error bound for stochastic MMD gradient in theorem 4.

The experimental part is very limited with few experiments. The methods is shown to work with simple data sets like MNIST and FashionMNIST, when they considered a much more complicated-structure data set like CIFAR10, then its FID is quite bad compared to NCSN and WGAN.

I do not find both theory and application part strong enough for publication.

**Questions:**

No question

---

> ### Author Response · Authors · 2023-11-16
> **Rebuttal**
>
> Thank you very much for the review.
>
> - We extended our experimental part by the CelebA dataset which admits more structure than (Fashion)MNIST.
> Here, our method admits very good results. In particular, we outperform WGAN significantly (32.1 vs 41.3 in the FID).
> Moreover, on CIFAR10 our method is at least competitive with other gradient-flow-based methods though we agree that there is a gap between gradient-flow-based and score-based methods like NCSN (in our paper as well as in the literature). Compared with WGAN, we obtain with 54.8 vs 55.2 a similar (even slightly better) FID.
> Note that previous works on MMD flows (e.g. Arbel et al, 2019) were not able to scale to image generation because of the quadratic computational cost within the number of samples. As mentioned by reviewer HuoH our fast gradient computation "opens up the possibility of employing MMD flows in generative modeling tasks that were previously considered impractical for these types of functions".
>
> - Regarding the theoretical part, we would like to emphasize that our results provably lower the computational complexity of (gradient) evaluations of the MMD/energy distance.
> All in all the proofs of our theorems are non-trivial and require a wide range of mathematical tools.
> They are directly related to the applications part and result in a highly efficient algorithm for MMD (gradient) computations.
>
> We hope that the reviewer will reconsider our paper with these explanations.
>
> References:
>
> (Arbel et al, 2019) Maximum Mean Discrepancy Gradient Flow

---

> > ### Comment · Reviewer_e9Tn · 2023-11-23
> > **Response to the rebuttal**
> >
> > Dear authors,
> >
> > I would like to thank the authors for their responses. I think we disagree respectfully about the level of difficulty of theory presented in paper as well as the scope of empirical work needed to be done to convince the community when introducing a new method. Thus, I keep my initial score.

---

> > > ### Author Response · Authors · 2023-11-23
> > > **Response**
> > >
> > > Thank you very much for your comments. We present an algorithm that provably speeds up the computation of (gradients of) the energy distance, which is heavily used in machine learning applications. Our algorithm is implementable within very few lines of code and we are not aware of any comparable method in the literature. The impact on the computation time is numerically verified in Fig 1 on the left.
> > >
> > > Therefore, we think that the level of difficulty of the proofs is not too important for our contribution, even though we disagree with the reviewers opinion that they would be trivial. It is not the purpose of a proof to be complicated or difficult.

---

### Author Response · Authors · 2023-11-16
**General answer**

We would like to thank all the reviewers for the evaluation of our manuscript.
We carefully updated the paper.
In order to demonstrate that our generative sliced MMD flows also work on higher dimensional image datasets, we added an experiment on CelebA. Additionally, we answered and addressed all questions and suggestions from the reviewers separately, directly below the corresponding review.

---

### Meta-Review · Area_Chair_49qN · 2023-12-07

**Metareview:**

The paper proposes an novel efficient way of computing MMD of Riesz kernels and its gradient.  The authors then
use  this metric for generative modellin through gradient flow.

While score of the paper are mixed, reviewers found that the paper (after discussions and rebuttal) is interesting and introducing novel ideas
that deserves to be presented at the conference

**Justification For Why Not Higher Score:**

The framework is very specific to a given kernel, limiting a bit the applicability of the method.

**Justification For Why Not Lower Score:**

the paper propose a novel and interesting idea for accelerating the computation of MMD

---

### Decision · Program_Chairs · 2024-01-16

Accept (poster)